# Mapping person-to-person variation in viral mutations that escape polyclonal serum targeting influenza hemagglutinin

Juhye M Lee[1,2], Rachel Eguia[1], Seth J Zost[3], Saket Choudhary[4], Patrick C Wilson[5], Trevor Bedford[6], Terry Stevens-Ayers[6], Michael Boeckh[6], Aeron C Hurt[7], Seema S Lakdawala[8], Scott E Hensley[3], Jesse D Bloom[1,2,9]*

[1]Basic Sciences Division, Fred Hutchinson Cancer Research Center, Seattle, United States; [2]Department of Genome Sciences, University of Washington, Seattle, United States; [3]Department of Microbiology, Perelman School of Medicine, University of Pennsylvania, Philadelphia, United States; [4]Department of Biological Sciences, University of Southern California, Los Angeles, United States; [5]Department of Medicine, Section of Rheumatology, University of Chicago, Chicago, United States; [6]Vaccine and Infectious Disease Division, Fred Hutchinson Cancer Research Center, Seattle, United States; [7]WHO Collaborating Centre for Reference and Research on Influenza, Peter Doherty Institute for Infection and Immunity, Melbourne, Australia; [8]Department of Microbiology and Molecular Genetics, School of Medicine, University of Pittsburgh, Pittsburgh, United States; [9]Howard Hughes Medical Institute, Seattle, United States

**Abstract** A longstanding question is how influenza virus evolves to escape human immunity, which is polyclonal and can target many distinct epitopes. Here, we map how all amino-acid mutations to influenza's major surface protein affect viral neutralization by polyclonal human sera. The serum of some individuals is so focused that it selects single mutations that reduce viral neutralization by over an order of magnitude. However, different viral mutations escape the sera of different individuals. This individual-to-individual variation in viral escape mutations is *not* present among ferrets that have been infected just once with a defined viral strain. Our results show how different single mutations help influenza virus escape the immunity of different members of the human population, a phenomenon that could shape viral evolution and disease susceptibility.
DOI: https://doi.org/10.7554/eLife.49324.001

*For correspondence:
jbloom@fredhutch.org

Competing interests: The authors declare that no competing interests exist.

## Introduction

Infection of humans with influenza virus elicits potent neutralizing antibodies targeting the viral hemagglutinin (HA) protein. These antibodies provide long-lasting immunity against the viral strain that elicited them (*Couch, 1975*; *Davies et al., 1982*; *Couch and Kasel, 1983*; *Yu et al., 2008*; *Krammer, 2019*). Unfortunately, the effectiveness of this immunity against future strains is rapidly degraded by viral antigenic evolution (*Bedford et al., 2014*; *Smith et al., 2004*), such that the typical human is infected by influenza virus roughly every 5 years (*Couch and Kasel, 1983*; *Kucharski et al., 2015*; *Ranjeva et al., 2019*).

Classic studies of this evolutionary process demonstrated that it is easy to experimentally select mutant viruses that escape neutralization by individual monoclonal antibodies (*Yewdell et al., 1979*; *Laver et al., 1979*; *Webster and Laver, 1980*; *Gerhard et al., 1981*). But these same studies also found that monoclonal antibodies target a variety of non-overlapping regions on HA, such that no

**eLife digest** The human immune system protects the body from repeat attacks by remembering past infections. However, a typical person comes down with the flu every five to seven years. This is because flu viruses rapidly evolve to bypass our defenses. So, after a few years, the viruses look so different that the immune system no longer recognizes them.

The immune system recognizes flu viruses by producing proteins known as antibodies, which can bind to the virus and prevent it from infecting cells. Many of these antibodies bind to a protein on the surface of the virus called hemagglutinin, but each anti-flu antibody recognizes only a small region of the protein. This means that to escape recognition by a single antibody, all the virus needs to do is wait for a lucky mutation to change the part of hemagglutinin recognized by that antibody. But humans make many different antibodies. To escape them all, flu viruses would need lots of lucky mutations. So how do flu viruses keep winning the evolutionary lottery?

To answer this question, Lee et al. made all the possible individual mutations to the hemagglutinin protein of a human flu virus. A pool of these viruses was then exposed to the full mix of antibodies present in human serum (the liquid component of blood). Lee et al. then checked which mutations helped the virus survive contact with the antibodies. For most human serum samples, a single mutation was enough to allow the virus to escape most of one person's anti-flu antibodies. This suggests that the immune response to flu is so focused on a small region of hemagglutinin that a mutation in this region can enable the virus to take a huge step towards evading immune detection.

Even more surprising was what happened when Lee et al. looked at serum from different people. A mutation that helped the virus to escape immune detection in one person often had little or no effect on escape from another person's immunity. In other words, the lucky mutation that the virus needed to escape differed from one person to the next.

Every year there are many related flu viruses that infect humans. The results of Lee et al. suggest that people could be susceptible to different forms of the virus. Understanding how flu viruses escape immune detection in different people could help us identify which version of the virus different people are more susceptible to, and perhaps eventually better predict how the virus will evolve and spread.

DOI: https://doi.org/10.7554/eLife.49324.002

single viral mutation can escape a mix of antibodies targeting distinct regions (*Yewdell et al., 1979*; *Laver et al., 1979*; *Webster and Laver, 1980*; *Caton et al., 1982*). This work therefore posed a perplexing question: given that human immunity is polyclonal, how does influenza virus evolve to escape all the myriad antibody specificities in human sera?

Two primary explanations have been suggested for this conundrum. One explanation proposes that polyclonal immunity simply selects mutations that provide generalized neutralization resistance by increasing receptor avidity, without necessarily abrogating antibody binding (*Yewdell et al., 1986*; *Hensley et al., 2009*). Another explanation proposes that human immunity is sufficiently focused on one epitope in HA that single viral mutations can appreciably reduce binding by the mix of antibodies in polyclonal sera. This latter explanation has been shown to be true for the immunity of certain individuals against H1N1 influenza virus (*Li et al., 2013*; *Linderman et al., 2014*; *Huang et al., 2015*; *Davis et al., 2018*). It is also supported by mass spectrometry studies showing that HA-binding antibodies after vaccination are often dominated by relatively few 'clonotypes' (*Lee et al., 2016*; *Lee et al., 2019*), and studies with ferret sera suggesting that mutations to a small number of HA residues near the receptor binding pocket can lead to large antigenic changes (*Koel et al., 2013*). Of course, how polyclonal immunity exerts selection on influenza virus may depend on the serum in question. However, although several studies have selected individual viral mutants with enhanced resistance to neutralization by human sera (*Davis et al., 2018*; *DeDiego et al., 2016*; *Li et al., 2016*), there have been no systematic analyses of the immune selection exerted by sera from different individuals.

Here, we harness mutational antigenic profiling (*Doud et al., 2017*; *Doud et al., 2018*) to map how all amino-acid mutations to HA affect neutralization of H3N2 influenza virus by human serum.

We show that human serum can select single mutations that reduce viral neutralization by over an order of magnitude. Although the escape mutations usually occur in a similar region on HA's globular head, there is remarkable person-to-person variation in their antigenic effect: we identify mutations that reduce neutralization by >10-fold for one individual's serum but have little effect for another individual's serum. Our work suggests that person-to-person variation in the fine specificity of anti-influenza immunity may play a major role in shaping viral evolution and disease susceptibility.

## Results

### Mutational antigenic profiling comprehensively maps immune-escape mutations

Prior work studying immune selection from polyclonal sera has used escape-mutant selections, which simply isolate individual mutant viruses with reduced neutralization sensitivity. As a more comprehensive alternative, we recently developed mutational antigenic profiling (*Doud et al., 2017*; *Doud et al., 2018*). As illustrated in *Figure 1*, this approach quantifies how every amino-acid mutation to HA affects viral neutralization. Here, we perform mutational antigenic profiling using mutant virus libraries with the HA from A/Perth/16/2009 (*Lee et al., 2018*), which was the H3N2 component of the influenza vaccine from 2010 to 2012 (*WHO, 2019*).

To validate mutational antigenic profiling for these viral libraries, we first performed selections with four monoclonal antibodies isolated from humans who had received the 2010–2011 trivalent influenza vaccine (*Zost et al., 2017*; *Henry Dunand et al., 2015*). These four antibodies were 037-10036-5A01, 034-10040-4C01, 028-10134-4F03, and 041-10047-1C04, hereafter referred to as '5A01', '4C01', '4F03', and '1C04', respectively. Based on binding assays to a small HA mutant panel, two antibodies target near the receptor-binding pocket, while two antibodies target lower on HA's head (*Zost et al., 2019*). For each antibody, we performed at least two replicates of mutational antigenic profiling using independently generated virus libraries and an antibody concentration such that <10% of the library retained infectivity after neutralization (*Figure 2—figure supplement 1*). Note that the magnitude of the measured immune selection depends on the strength of antibody selection (*Doud et al., 2017*), so while heights of letters in our maps of immune selection (i.e. logo plots like the one at right of *Figure 1*) can be compared within a given map, y-axis scales are not directly comparable across maps.

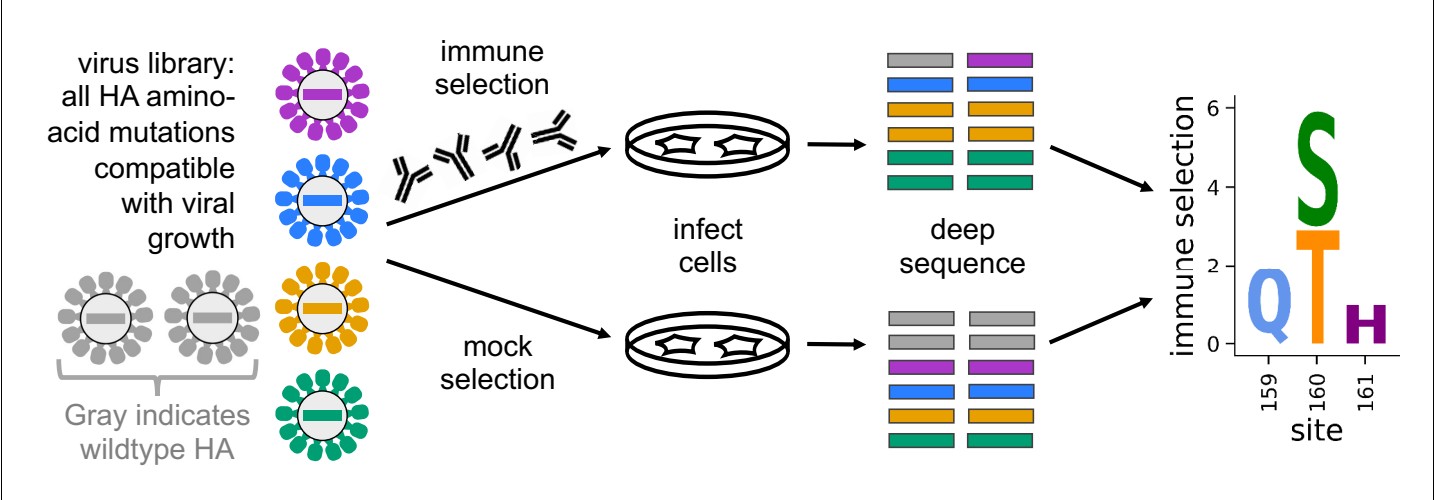

**Figure 1.** Mutational antigenic profiling quantifies the antigenic effect of all amino-acid mutations to HA. We generate libraries of mutant viruses carrying all mutations to HA compatible with viral growth. We incubate the libraries with antibodies or serum, and infect cells with non-neutralized virus. Viral RNA from infected cells is deep sequenced to measure the frequency of each mutation. We quantify immune selection on each mutation as its log enrichment relative to wildtype in the immune-selected sample versus a mock-selection control. Data are displayed in logo plots, with larger letters indicating stronger immune selection for a mutation.
DOI: https://doi.org/10.7554/eLife.49324.003

Importantly, the selected mutations were consistent across the biological replicates with independently generated mutant virus libraries (*Figure 2—figure supplement 2*), confirming that our approach systematically maps the antigenic effects of all mutations rather than simply selecting one-off viral mutants. The strongly selected mutations occurred mostly at sites that are classically categorized as antigenic region B (*Supplementary file 1*; *Wiley et al., 1981*; *Wu and Wilson, 2017*). However, although the antibodies all targeted the same antigenic region of HA, they selected different escape mutations (*Figure 2A*). These antibody-to-antibody differences in escape mutations were validated by traditional neutralization assays (*Figure 2B*). For instance, K160T reduces neutralization by antibody 5A01 but has no effect on neutralization by antibody 4C01 despite the fact that both antibodies target a similar region of HA (*Figure 2E*).

The two antibodies that we expected to target lower on HA's head indeed selected mutations in this portion of HA, mostly at sites classically categorized as antigenic regions C, D, and E (*Supplementary file 1*). But again the specific mutations selected by each antibody differed (*Figure 2C,D,F*). Overall, these results demonstrate that mutational antigenic profiling accurately maps antibody selection on the Perth/2009 HA, and underscore the observation that different antibodies targeting an apparently similar epitope often have distinct escape mutations (*Doud et al., 2017*; *Dingens et al., 2019*).

## Contemporary human sera strongly selects specific escape mutations

We next mapped immune selection from polyclonal human sera. We reasoned that it would be most informative to examine serum collected during the time when the Perth/2009 virus was circulating in the human population. We therefore screened the neutralizing activity of sera collected from 16 healthy adults between 2008 and 2010, and identified four sera that completely neutralized virus with the wild-type Perth/2009 HA at a dilution of $\geq$1:40 (which is the HAI titer traditionally assumed to reduce risk of infection; *Hobson et al., 1972*). Notably, this pre-screening means that all the sera that we characterized had high neutralizing activity, with inhibitory concentrations 50% (IC50s) ranging from ~1:800 to ~1:3000 (*Supplementary file 2*). No information on influenza vaccination or infection history was available for these four individuals, who ranged in age from 21 to 65 years old at the time of collection. We performed mutational antigenic profiling using serum dilutions chosen so that ~5% of the mutant virus library survived neutralization, although the exact percentage varied slightly among sera and replicates (*Figure 3—figure supplement 1*). We targeted this amount of neutralization to impose strong selection while still allowing identification of mutations that might mediate only partial escape.

Despite being polyclonal, each serum strongly selected single escape mutations (*Figure 3A*). The selected mutations occurred at a relatively small number of sites, predominantly in the portion of HA classically categorized as antigenic region B (*Figure 3A,C* and *Supplementary file 1*). The sites of selection were reproducible across three biological replicates using independently generated mutant virus libraries (*Figure 3—figure supplement 2*).

To validate the mutations identified by the antigenic profiling, we performed neutralization assays on viruses carrying individual mutations (*Figure 3B*). In all cases, the most strongly selected mutation in the profiling had a large antigenic effect in a neutralization assay. For three individuals (the 21-year-old, 64-year-old, and 65-year-old), the strongest escape mutation had a >10-fold effect on neutralization, and for the remaining individual (the 53-year-old) the strongest escape mutation had a ~5-fold effect (*Figure 3B*). Amazingly, the antigenic effect of the strongest escape mutant from three of the polyclonal sera was comparable to that of strongest escape mutant from the monoclonal antibody 5A01 (compare *Figure 2B* to *Figure 3B*). For most of the sera, the strongest escape mutation completely escaped neutralization at the serum concentration used in the mutational antigenic profiling (see dashed lines in *Figure 3B*). These results show that single mutations can dramatically reduce viral sensitivity to neutralization by polyclonal human sera.

However, the exact mutations that mediated escape differed markedly across sera (*Figure 3A*). For instance, F193D escaped viral neutralization by the serum from the 21-year-old individual by >10-fold—but had no effect for the serum from the 64-year-old (*Figure 3B*). Similarly, F159G escaped neutralization by the 64-year-old's serum but had minimal effect on neutralization by the 53-year-old (*Figure 3B*). *Figure 3A,B* shows numerous other examples of such person-to-person variation in viral escape mutations. In fact, the only viral mutation that had a consistent antigenic effect across individuals is K160T, which moderately enhanced neutralization resistance to all four sera.

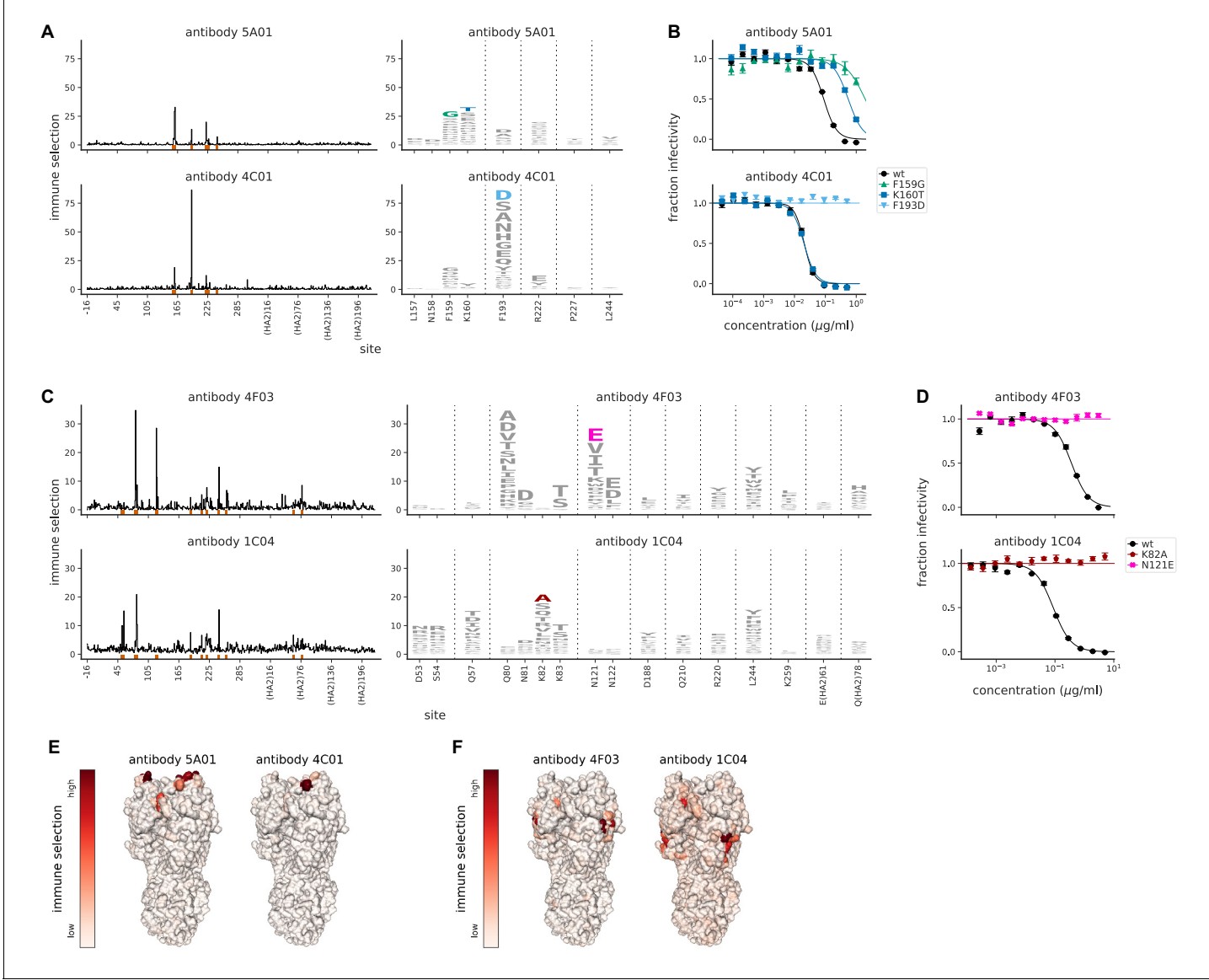

**Figure 2.** Mutational antigenic profiling of monoclonal antibodies targeting HA. (A) Maps of immune selection from two antibodies targeting near the receptor binding pocket. Line plots show the total immune selection at each site, and logo plots show mutations at strongly selected sites (indicated by red marks at the bottom of the line plots). (B) Neutralization assays validate that mutations mapped to be under strong immune selection indeed have large effects on neutralization. For each antibody, we validated the mutation under strongest selection (F159G for 5A01 and F139D for 4C01), as well as K160T since it was selected by 5A01 but not 4C01. (C) Maps of immune selection for two antibodies targeting lower on the HA head and (D) corresponding neutralization curves for the most strongly selected escape mutation for each antibody (N121E for 4F03 and K82A for 1C04). (E) Immune selection mapped onto HA's structure for the antibodies targeting near the receptor binding pocket or (F) lower on the HA head. The color scale is calibrated separately for each structure. For interactive versions of the structures, click https://mybinder.org/v2/gh/jbloomlab/map_flu_serum_Perth2009_H3_HA/master?urlpath=%2Fapps%2Fresults%2Fnotebooks%2Fmap_on_struct_antibody_region_B.ipynb and https://mybinder.org/v2/gh/jbloomlab/map_flu_serum_Perth2009_H3_HA/master?urlpath=%2Fapps%2Fresults%2Fnotebooks%2Fmap_on_struct_antibody_lower_head.ipynb.

DOI: https://doi.org/10.7554/eLife.49324.004

The following figure supplements are available for figure 2:

**Figure supplement 1.** Percent of viral library retaining infectivity after antibody treatment during mutational antigenic profiling.

DOI: https://doi.org/10.7554/eLife.49324.005

**Figure supplement 2.** Biological replicates of the mutational antigenic profiling are well correlated.

DOI: https://doi.org/10.7554/eLife.49324.006

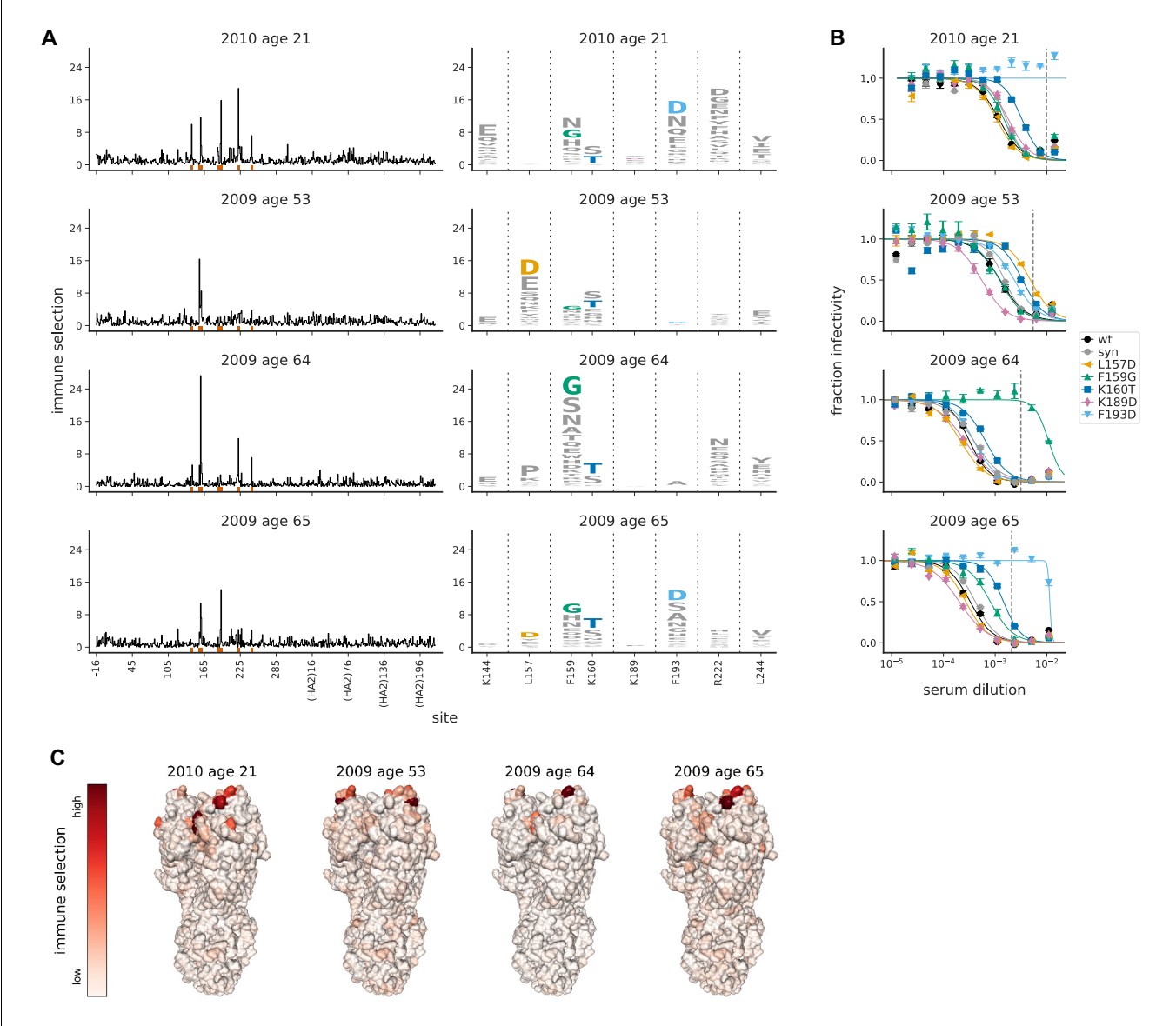

**Figure 3.** Mutational antigenic profiling of four human serum samples. Plot titles indicate the year the serum was collected and the age of the individual at that time. (**A**) Line plots show the total immune selection at each site, and logo plots show mutations at strongly selected sites. (**B**) Neutralization assays validate that mutations mapped to be under strong immune selection indeed have large antigenic effects. The dashed vertical lines show the serum concentrations used in the mutational antigenic profiling. We validated five amino-acid mutants against all four sera, choosing mutations for the following reasons: F193D is the most strongly selected mutation for the 21- and 65-year-old, L157D is the most selected for the 53-year-old, F159G is the most selected for the 64-year-old, K160T is modestly selected by all sera, and K189D is strongly selected by ferret sera (see below). The 'syn' mutant is a control with a synonymous mutation does not change the protein sequence and so should not affect antigenicity. (**C**) Immune selection mapped onto HA's structure for each sera. The color scale is calibrated separately for each structure. For interactive versions of the structures, click https://mybinder.org/v2/gh/jbloomlab/map_flu_serum_Perth2009_H3_HA/master?urlpath=%2Fapps%2Fresults%2Fnotebooks%2Fmap_on_struct_VIDD_sera.ipynb. Site 189 is shown in the logo plots despite not being a strongly selected site because it is important for ferret sera (see below).

DOI: https://doi.org/10.7554/eLife.49324.007

The following figure supplements are available for figure 3:

**Figure supplement 1.** Percent of viral library retaining infectivity after serum treatment during mutational antigenic profiling.
DOI: https://doi.org/10.7554/eLife.49324.008

**Figure supplement 2.** Biological replicates of the mutational antigenic profiling are well correlated.
DOI: https://doi.org/10.7554/eLife.49324.009

This serum-to-serum variation suggests that we are mapping serum-specific antigenic mutations rather than avidity mutants that generally enhance neutralization resistance (*Yewdell et al., 1986*; *Hensley et al., 2009*). It also shows that serum-escape mutations in the same classically defined antigenic region can have widely different effects across individuals.

To confirm that our serum-escape maps are replicable over short timeframes during which there are not expected to be large changes in underlying immunity, we examined a second serum sample from the 53-year-old individual collected two months after the sample mapped in *Figure 3*. There is no indication that the individual was infected or vaccinated during these 2 months (which spanned from January to March of 2009), and an absence of immune exposure is further supported by the fact that there was no rise in serum neutralizing titer during this time (*Supplementary file 2*). *Figure 4* shows that the maps of immune selection were highly similar for the two samples, consistent with the expectation that the specificity of the serum was stable over the 2 months.

## Mapping viral escape mutations pre- and post-vaccination

Because vaccination can shape anti-influenza serum responses (*Fonville et al., 2014*; *Zost et al., 2017*; *Levine et al., 2019*), we next performed mutational antigenic profiling with sera collected from four individuals pre-vaccination and 1 month post-vaccination. These individuals, who ranged in age from 25 to 49 years, received the 2015–2016 vaccine, for which the H3N2 component was A/Switzerland/9715293/2013 (*WHO, 2019*). Because these sera were collected 6 years after 2009 from individuals vaccinated with an antigenic successor of Perth/2009, we anticipated that it might be more difficult to select escape mutations in our Perth/2009 library, since the sera is likely to have immunity both to Perth/2009-like viruses and their naturally occurring antigenic drift variants.

The serum from one individual (the 25-year-old) strongly selected escape mutants in the Perth/2009 library even pre-vaccination (*Figure 5A*). This individual's serum also had the most potent pre-vaccination neutralizing activity against Perth/2009, with an IC50 of ~1:800 (*Figure 5B*). The mutation

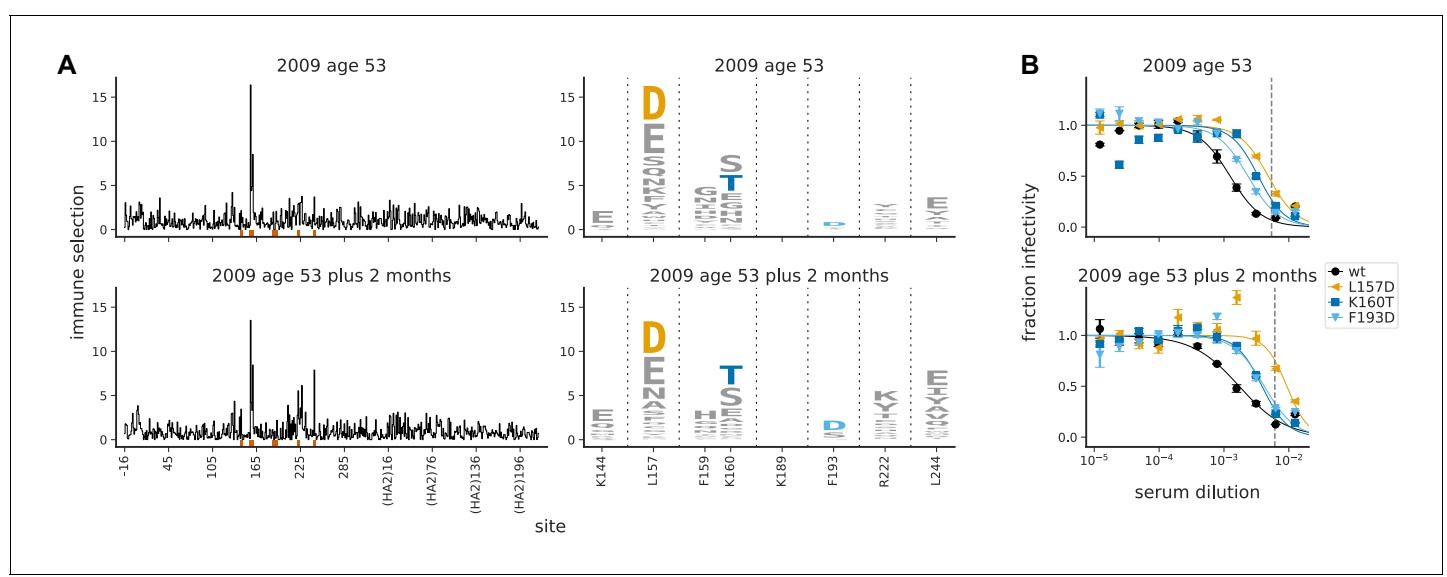

**Figure 4.** The maps of immune selection are stable over short time periods in the absence of vaccination or infection. This figure re-displays the (A) map of immune selection and (B) neutralization curves from *Figure 3* for the serum from the 53-year-old individual alongside comparable data generated using another serum sample from the same individual collected 2 months later. The dashed vertical lines on the neutralization curves show the serum concentrations used in the mutational antigenic profiling. The mutations chosen for validation by neutralization curves are the ones that the neutralization curves in *Figure 3B* found had an effect for sera from this individual.

DOI: https://doi.org/10.7554/eLife.49324.010

The following figure supplements are available for figure 4:

**Figure supplement 1.** Percent of viral library retaining infectivity after serum treatment during mutational antigenic profiling.
DOI: https://doi.org/10.7554/eLife.49324.011

**Figure supplement 2.** Biological replicates of the mutational antigenic profiling are well correlated.
DOI: https://doi.org/10.7554/eLife.49324.012

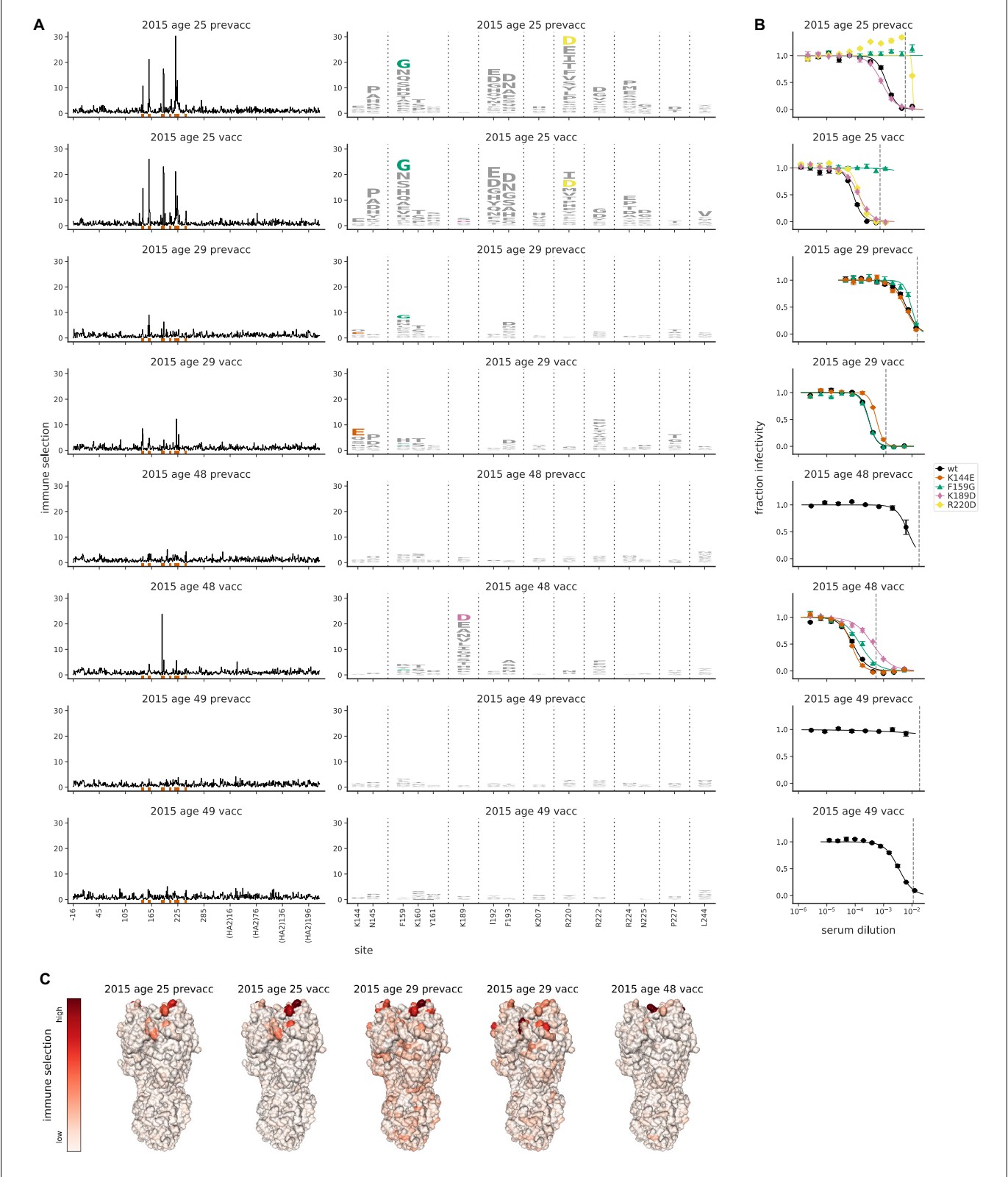

**Figure 5.** Mutational antigenic profiling of sera from four humans pre- and post-vaccination. Plot titles indicate the year the serum was collected, the age of the individual at that time, and the vaccination status. (**A**) Line plots show total immune selection at each site, and logo plots show mutations at strongly selected sites. (**B**) Neutralization assays validate that mutations mapped to be under strong immune selection indeed have large antigenic effects. The dashed vertical lines show the serum concentrations used in the mutational antigenic profiling. Note that not all mutations are tested

*Figure 5 continued on next page*

*Figure 5 continued*

against all sera. (C) Immune selection mapped onto HA's structure for sera that selected escape mutants. The color scale is calibrated separately for each structure. For interactive versions of the structures, click https://mybinder.org/v2/gh/jbloomlab/map_flu_serum_Perth2009_H3_HA/master? urlpath=%2Fapps%2Fresults%2Fnotebooks%2Fmap_on_struct_Hensley_sera.ipynb.

DOI: https://doi.org/10.7554/eLife.49324.013

The following figure supplements are available for figure 5:

**Figure supplement 1.** Percent of viral library retaining infectivity after serum treatment during mutational antigenic profiling.

DOI: https://doi.org/10.7554/eLife.49324.014

**Figure supplement 2.** Biological replicates of the mutational antigenic profiling are well correlated.

DOI: https://doi.org/10.7554/eLife.49324.015

most strongly selected by the pre-vaccination serum (F159G) reduced neutralization by over an order of magnitude (*Figure 5B*). Vaccination of this individual substantially enhanced the serum's potency, dropping the IC50 to ~1:10,000—but F159G continued to have the largest effect on neutralization, still reducing the IC50 by over an order of magnitude (*Figure 5A,B*).

The second most potent pre-vaccination serum (that of the 29-year-old, which had an IC50 of ~1:150) also perceptibly selected an antigenic mutation (F159G), but the effect of the mutation was much smaller (*Figure 5A,B*). Vaccination enhanced the serum potency by ~20-fold, and also slightly shifted its specificity (*Figure 5A,B*). Prior to vaccination, F159G but not K144E had a small antigenic effect, whereas the finding was reversed post-vaccination. Notably, both F159 and K144 are mutated in the Switzerland/2013 virus relative to Perth/2009. However, the fact that no mutations greatly alter the IC50 of this individual's serum suggest that it is not narrowly focused on any specific epitope on the Perth/2009 HA.

The remaining two individuals (the 48- and 49-year-old) had low serum potency before vaccination (*Figure 5B*), such that we were unable to exert sufficient immune selection to map escape mutations from their pre-vaccination sera (*Figure 5A* and *Figure 5—figure supplement 1*). Vaccination enhanced the serum potency of both individuals (*Figure 5B*). For the 48-year-old, this enhanced neutralizing activity was strongly focused, with the K189D mutation decreasing the IC50 of the post-vaccination serum by an order of magnitude (*Figure 5A,B*). But for the 49-year-old, the enhanced neutralizing activity was not associated with increased selection of any HA mutations (*Figure 5A*), suggesting that either the vaccine response was not dominated by any single HA specificity, or that it targeted epitopes that are not tolerant of viral escape mutations.

These results show that it is possible to select strong escape mutants even from serum collected well after the Perth/2009 virus circulated from individuals vaccinated with an antigenic successor of this virus. In some cases vaccination shifts the specificity of the serum, whereas in other cases it primarily boosts existing specificities, consistent with prior work using other approaches (*Fonville et al., 2014*; *Lee et al., 2016*; *Lee et al., 2019*; *Henry et al., 2019*). As in the previous section, the strong escape mutations are predominantly but not exclusively in antigenic region B of HA (*Figure 5C* and *Supplementary file 1*)—but again the effects of specific mutations vary markedly among sera.

## Immune selection by sera from singly infected ferrets is consistent and different from human sera

The antigenicity of influenza viral strains is currently characterized mostly using sera from ferrets that have been infected just once with a single viral strain (*Smith et al., 2004*). In contrast, humans have complex exposure histories that may influence their antibody response (*Li et al., 2013*; *Linderman et al., 2014*; *Andrews et al., 2015b*; *Gostic et al., 2016*; *Cobey and Hensley, 2017*). To compare selection from the sera of humans and singly-infected ferrets, we performed mutational antigenic profiling using sera from five ferrets. Three ferrets were infected with the Perth/2009 viral strain at the University of Pittsburgh. The other two ferrets were infected at the World Health Organization (WHO) Collaborating Centre in Melbourne, one with Perth/2009 and one with A/Victoria/361/2011 (the immediate antigenic successor of Perth/2009 in the influenza vaccine; *WHO, 2019*). We used ferret sera from different labs and viral strains to sample across factors that might affect serum specificity.

In stark contrast to the person-to-person variation of the human sera, the maps of immune selection were very similar for all post-infection ferret sera (*Figure 6A*). This was true even for the ferret infected with Victoria/2011 rather than Perth/2009 virus: visual comparison of *Figure 6A* and *Figure 3A* suggests that the differences between ferrets infected with these two antigenically distinct H3N2 viruses are smaller than the differences among nearly all the human sera. To statistically confirm this observation, we calculated the beta diversity of the site-level selection for each group of serum using the Simpson index (*Jost, 2007*). Higher beta diversity indicates that the sera in a group show more variation in the HA sites where they exert selection. This analysis confirmed that the ferret sera (beta diversity of 1.12) were more consistent in the sites where they exerted selection than the human sera from either 2009–2010 or post-vaccination from 2015 (beta diversity of 1.38 and 1.60, respectively).

The strongest selection from the ferret sera focused on sites 189 and 193, with mutations K189D and F193D tending to have a ~5 to 10-fold effect on neutralization (*Figure 6B*). Prior work has also noted that these two sites are important for antigenic recognition by ferret sera (*Koel et al., 2013*). Mutations at a handful of additional sites (such as 142, 144, and 222) were also modestly selected by some ferret sera (*Figure 6A,B*).

Although the ferret sera was similar to the human sera in focusing mostly on antigenic region B of HA (*Figure 6C* and *Supplementary file 1*), there were notable differences in the specific sites where mutations were selected. The site where mutations were most strongly selected by all the ferret sera (site 189) had an appreciable antigenic effect for only one of the human sera. The other site where mutations were consistently and strongly selected by the ferret sera (site 193) had an antigenic effect for less than half the human sera. The converse was also true: many sites of mutations strongly selected by human sera (such as 157, 159, and 160) had little antigenic effect for the ferret sera. These findings highlight major differences in the fine specificity of immune focusing between ferrets that have been infected with a single viral strain and humans with complex exposure histories.

## The effect of a monoclonal antibody in polyclonal serum

The preceding sections show that polyclonal sera often select single antigenic mutations—a phenomenon more typically associated with monoclonal antibodies. To test how a single antibody can contribute to the selection of escape mutations by polyclonal serum, we spiked an antibody that targets lower on HA's head into serum that selects mutations at the top of HA's head. We spiked the antibody into the serum at three concentrations. In all cases, we kept the serum at a concentration where ~4% of the viral library survived neutralization by serum alone. The three antibody concentrations were chosen such that in the presence of antibody alone, the percents of the viral library that survived neutralization were ~25% (low concentration), ~5% (mid concentration), and ~2% (high concentration). In the actual selections with the mix of serum and antibody, there was combined selection from the serum and antibody, so that the percents of the library that survived neutralization were lower than those due to the serum or antibody alone. Specifically, the percents of the library that survived selection with the mix were 1.7% (low antibody), 0.16% (mid antibody), and 0.015% (high antibody); see *Figure 7—figure supplement 1* for details. These selections therefore span a range in which the antibody is a modest versus dominant contributor to the overall neutralizing activity of the mix.

Even when the serum was spiked with the lowest concentration of antibody, the mutational antigenic profiling showed signals of selection at sites targeted by the antibody (*Figure 7A*). At the middle antibody concentration, selection from the antibody exceeded that from the serum, although both were still apparent (*Figure 7A*). At the highest antibody concentration, selection at antibody-targeted sites completely overwhelmed selection at serum-targeted sites. These results do not imply a loss of serum activity when antibody is added, but rather indicate that at high concentration the antibody completely dominates the neutralizing activity of the mix, and so exerts all the measurable immune selection. But the dominance of a few sites in one of our antigenic maps does not imply that there are no antibodies targeting other sites—once a mutation escapes the dominant antibody specificity, mutations at other more weakly targeted sites would begin to have measurable antigenic effects. In other words, our maps show the *relative* effects of different mutations on viral neutralization, not the absolute effect of any given mutation on antibody binding.

In addition, there could be non-additive effects if antibodies sterically hinder each other or mutations induce conformational changes in HA. Part of the reason the antibody so readily dominates the

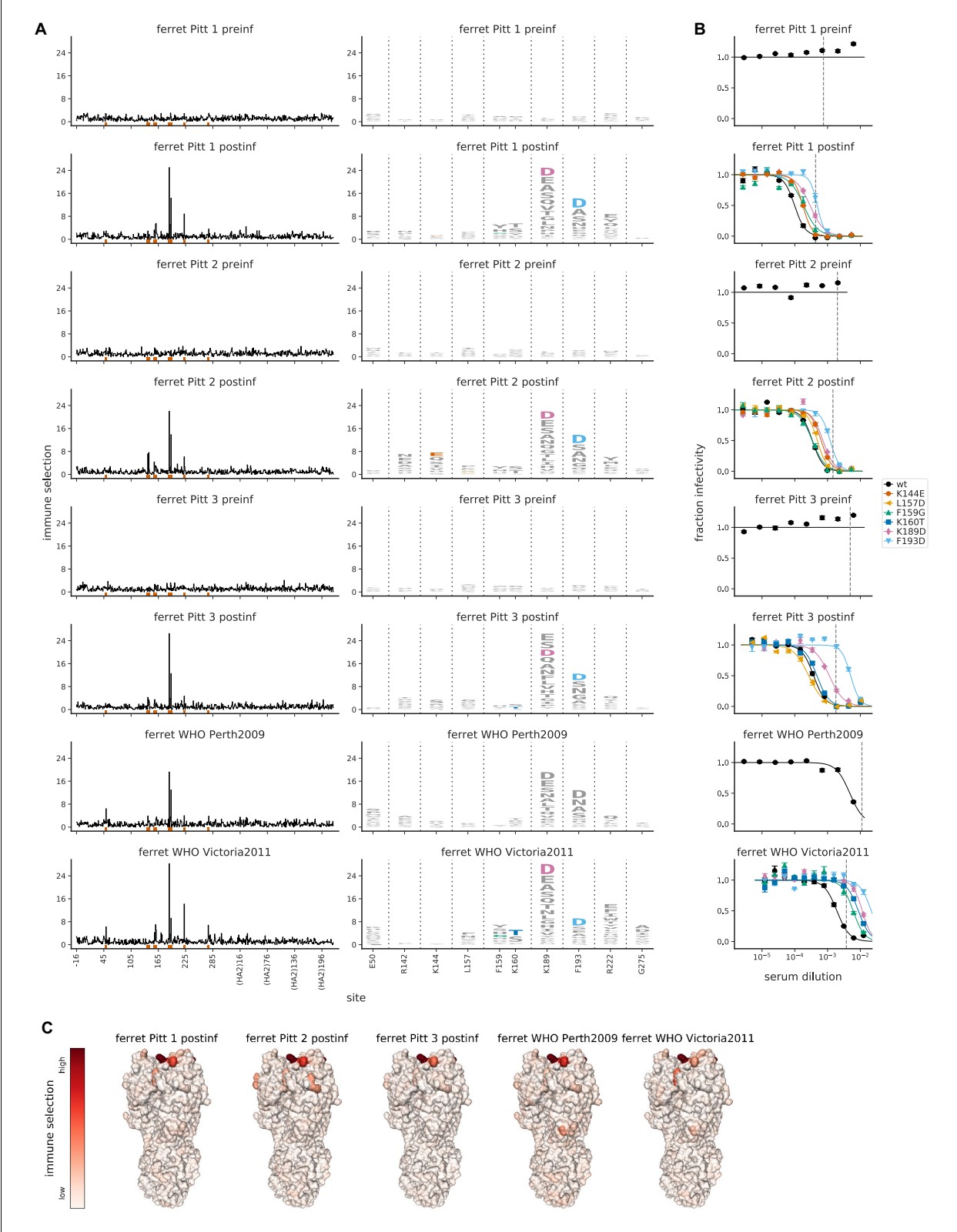

**Figure 6.** Mutational antigenic profiling of sera from five ferrets. Plot titles indicate the lab that performed the infection and if the infecting strain was Victoria/2011 rather than Perth/2009. For ferrets from Pittsburgh, both pre- and post-infection sera were analyzed as indicated in the plot titles. (**A**) Line plots show total immune selection at each site, and logo plots show mutations at strongly selected sites. (**B**) Neutralization assays validate that mutations mapped to be under strong immune selection indeed have large antigenic effects. The dashed vertical lines show the serum concentrations

*Figure 6 continued on next page*

*Figure 6 continued*

used in the mutational antigenic profiling. Note that not all mutations are tested against all sera. (C) Immune selection mapped onto HA's structure for sera that selected escape mutants. The color scale is calibrated separately for each structure. For interactive versions of the structures, click https://mybinder.org/v2/gh/jbloomlab/map_flu_serum_Perth2009_H3_HA/master?urlpath=%2Fapps%2Fresults%2Fnotebooks%2Fmap_on_struct_ferret.ipynb.
DOI: https://doi.org/10.7554/eLife.49324.016

The following figure supplements are available for figure 6:

**Figure supplement 1.** Percent of viral library retaining infectivity after serum treatment during mutational antigenic profiling.
DOI: https://doi.org/10.7554/eLife.49324.017

**Figure supplement 2.** Biological replicates of the mutational antigenic profiling are well correlated.
DOI: https://doi.org/10.7554/eLife.49324.018

selection in *Figure 7A* may be that the strongest serum-escape mutations (F159G and F193D) actually sensitize the virus to neutralization by the antibody (*Figure 7B*), a result reminiscent of earlier work showing synergistic effects of mixing antibodies (*Lubeck and Gerhard, 1982*; *Webster and Laver, 1980*). The loss of antibody selection at certain sites when serum is present (e.g. 220 and 259 in *Figure 7A*) also suggest possible synergistic effects of mixing antibody and serum. Overall, these spike-in experiments show that a single antibody can dominate selection if it comprises half or more of a serum's overall potency. Of course, these experiments cannot reveal if the focused selection from human sera is actually driven by a single antibody versus a collection of antibodies targeting a similar epitope.

## Many sites strongly selected by human sera change during natural evolution

If our mutational antigenic profiling reflects real antigenic pressure on human influenza virus, then we would expect to observe changes at sites in HA that our experiments mapped as being under selection. To test if this is true, we examined the natural evolution of human H3N2 influenza HA since 2007 at sites with clear signals of antigenic selection from at least one human serum (these are the 16 sites shown in the logo plots of *Figure 3A* and *Figure 5A*).

There has been substantial amino-acid evolution (new variants reaching $\geq$ 5% frequency) at nine of the sites strongly selected by the human sera (*Figure 8*). Across all sera that strongly selected mutations, the single largest-effect mutations occurred at five sites: 144, 157, 159, 189, and 193 (*Figure 3A* and *Figure 5A*). During the evolution of human H3N2 influenza virus, new amino-acid variants have reached >10% frequency at all these sites (*Figure 8*). Notably mutations at each of these five sites are selected by only some of the human sera that we examined.

The one HA mutation with an antigenic effect across almost all human sera we examined was K160T, which consistently caused a moderate increase in neutralization resistance (*Figure 3A,B*). This mutation introduces a N-linked glycosylation motif near the top of HA's head. K160T appeared among natural human H3N2 viruses in late 2013 and is now present in the majority of circulating viruses (*Figure 8*). In addition, K160T and mutations at three of the sites strongly selected by just some sera (144, 159, and 193) distinguish the 3C2.A and 3C3.A clades that currently circulate in humans (*Bedford and Neher, 2018*). Although the number of sera that we have characterized is too small to draw conclusions about population-level immunity, it is likely that evolutionary selection on a viral mutation depends both on how many individuals have immunity that is affected by the mutation and the magnitude of the antigenic effect in those individuals.

Notably, the amino-acid that emerges at a site in nature is not always the one that our experiments map as being under the strongest immune selection. There are several possible reasons: First, our experiments probe all amino-acid mutations but natural evolution is mostly limited to single-nucleotide changes. For instance, L157D is the largest-effect mutation at site 157 (*Figure 3*) but L157S appears in nature (*Figure 8*)—likely because S but not D is accessible from L by a single-nucleotide change. Second, natural evolution selects for viral replication and transmission as well as immune escape. Finally, the context of a mutation may influence its effect: for instance, F159Y has spread in nature but was not selected in our experiments, perhaps because in nature this mutation co-occurred with secondary changes such as N225D (*Chambers et al., 2015*; *Jorquera et al., 2019*). Importantly, it is now feasible to also make high-throughput measurements of how mutations to H3N2 HA affect viral replication (*Lee et al., 2018*) and epistatic interactions among sites

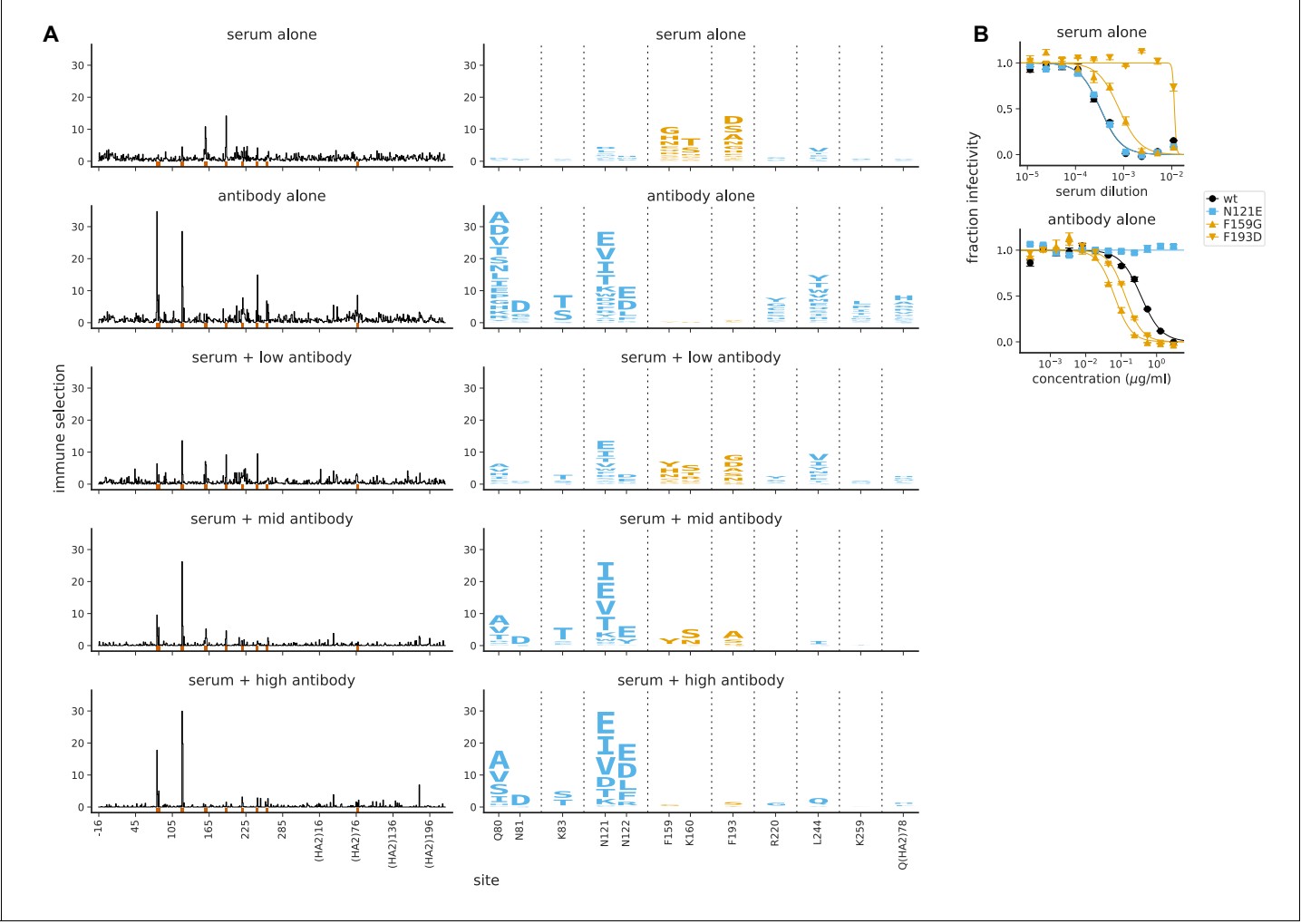

**Figure 7.** Mutational antigenic profiling of polyclonal human serum spiked with a monoclonal antibody. The antibody is spiked into the serum at a 'low' concentration (antibody alone less potent than serum alone), a 'mid' concentration (antibody similarly potent to serum), and a 'high' concentration (antibody more potent than serum); see *Figure 7—figure supplement 1* for details. (**A**) Line plots show the total immune selection at each site, and logo plots show mutations at sites that are strongly selected by the serum (orange) or antibody (blue). (**B**) Neutralization curves for some of the most strongly selected mutations against serum or antibody alone. Note that the antibody is 4F03 from *Figure 2*, the human serum is that from the 65-year-old in *Figure 3*, and the antibody-alone and serum-alone panels re-display data from those figures using a different color-scheme and subset of sites in the logo plots.

DOI: https://doi.org/10.7554/eLife.49324.019

The following figure supplements are available for figure 7:

**Figure supplement 1.** Percent of viral library retaining infectivity after treatment with each serum+antibody mix.

DOI: https://doi.org/10.7554/eLife.49324.020

**Figure supplement 2.** Biological replicates of the mutational antigenic profiling are well correlated.

DOI: https://doi.org/10.7554/eLife.49324.021

(*Wu et al., 2018*), so it may eventually be possible to build evolutionary models that incorporate all these factors (*Luksza and Lässig, 2014*; *Neher et al., 2016*).

## Discussion

We have mapped the selection that polyclonal human sera exert on all amino-acid mutations to a H3N2 influenza virus HA. Many sera are highly focused, and select single viral mutations that reduce neutralization by over an order of magnitude. This basic finding that human serum can be focused is

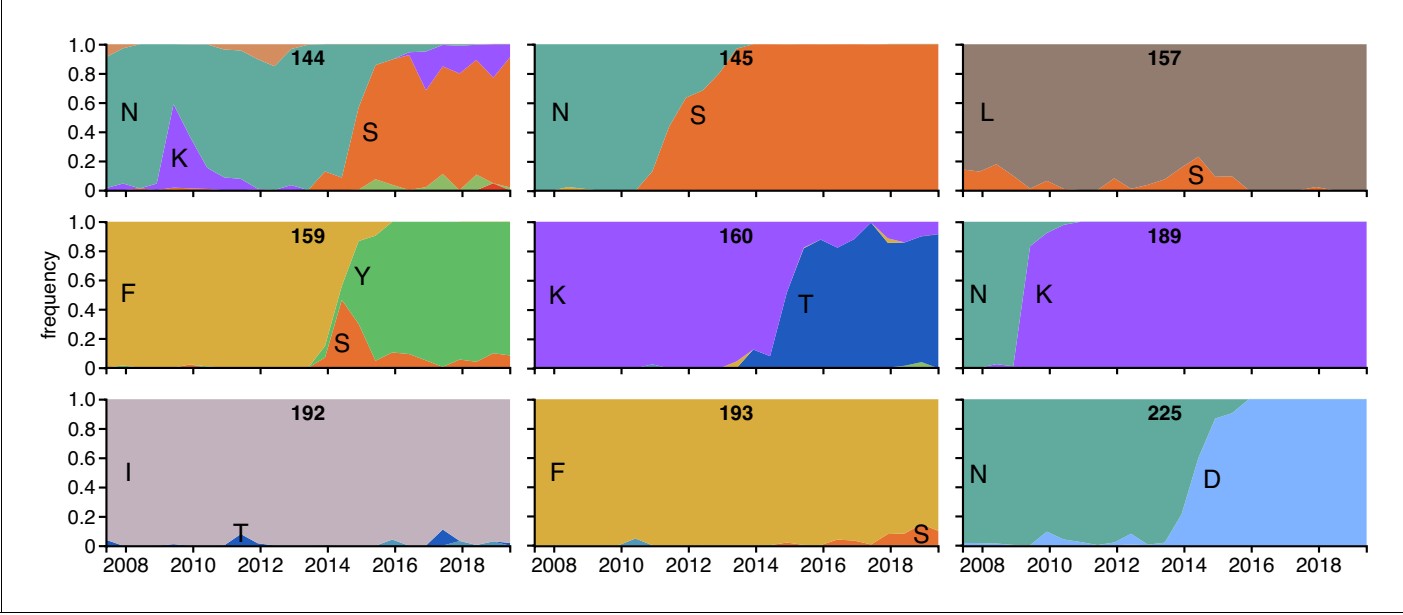

**Figure 8.** Frequencies of amino acids at key sites in human H3N2 influenza HA between 2007 and 2019. There are 16 sites under strong immune selection in our mutational antigenic profiling with human sera, and this figure shows the nine of these sites for which a new amino acid rose to ≥5% frequency. The numbers at the top of each plot indicate the site number, and the letters indicate the amino acid identity. The figure panels are taken from the Nextstrain real-time pathogen evolution website (*Hadfield et al., 2018*; *Neher and Bedford, 2015*).

DOI: https://doi.org/10.7554/eLife.49324.022

consistent with other recent work (*Li et al., 2013*; *Linderman et al., 2014*; *Huang et al., 2015*; *Davis et al., 2018*). But because our experiments mapped selection on all mutations, we were able to systematically quantify the extent of focusing and compare its targets across sera.

A striking result is that the targets of serum selection vary widely from person to person. Viral mutations that greatly reduce neutralization by one individual's serum sometimes have no effect for another individual's serum, which is instead affected by different mutations. Our results suggest a model in which a virus with a single mutation could be strongly favored in pockets of the human population with immunity focused on that site. Does this type of selection actually drive influenza virus evolution? Many of the mutations mapped in our experiments are at sites that have recently substituted in nature, suggesting that selection from focused human sera may be relevant to viral antigenic drift. However, it is important to note that we performed our selections using near-neutralizing concentrations of relatively potent human sera. There is evidence that sub-neutralizing serum concentrations more often select generalized resistance mutations that increase receptor avidity (*Yewdell et al., 1986*; *Hensley et al., 2009*). Both types of selection may occur in nature, and understanding their roles in shaping influenza virus evolution will require comparing the serum focusing of individuals with the viral strains that actually infect them.

The sera that we profiled predominantly focus on the portion of HA classically categorized as antigenic region B, consistent with other studies reporting that this region is immunodominant in recent H3N2 strains (*Popova et al., 2012*; *Chambers et al., 2015*; *Broecker et al., 2018*). However, our results highlight the limitations of subdividing HA into broad antigenic regions. These regions were originally defined based on the idea that a mutation selected by one antibody targeting a region often also abrogated binding of other antibodies targeting that region (*Webster and Laver, 1980*; *Wiley et al., 1981*; *Caton et al., 1982*). But our results show that different mutations in the same antigenic region can have very different effects across sera. The specificity of human sera is therefore more fine-grained than the classical categorization of HA into antigenic regions.

It is important to note that our experiments only mapped in vitro immune selection exerted by neutralizing anti-HA antibodies in serum. During actual human infections, antibodies to HA can also exert selection by non-neutralizing mechanisms such as antibody-dependent cellular cytoxicity (*He et al., 2016*; *Vanderven et al., 2018*), activation of complement (*Rattan et al., 2017*), and

inhibition of the viral neuraminidase (*Kosik et al., 2019*; *Chen et al., 2019*). In addition, antibodies in serum are mostly of the IgG class, but IgA antibodies predominate in the upper airway mucosa where human influenza virus typically replicates (*Gould et al., 2017*). But despite these caveats, many studies have shown that neutralizing serum activity is a strong correlate of protection against influenza (*Hobson et al., 1972*; *Wagner et al., 1987*; *Krammer, 2019*), so mapping selection from such serum provides an important if incomplete picture of the selection that human immunity imposes on the virus.

Why is the neutralizing activity of human polyclonal sera often so focused? One possibility is that very few distinct antibodies are responsible for the anti-HA activity of any given serum. Indeed, multiple studies have shown that relatively few clonotypes comprise the majority of the anti-HA serum repertoire of a single individual (*Andrews et al., 2015a*; *Lee et al., 2016*; *Lee et al., 2019*). Our findings further suggest that either most of these clonotypes target the same sites in HA, or that the serum's neutralizing activity is due to very few of the clonotypes. In support of the second idea, recent work has shown that as an individual ages, much of the antibody response is diverted to non-neutralizing epitopes because sites targeted by potently neutralizing antibodies change during viral evolution, while non-neutralizing epitopes are more often conserved (*Henry et al., 2019*; *Ranjeva et al., 2019*).

Prior work has shown that exposure history shapes human immunity to influenza (*Li et al., 2013*; *Andrews et al., 2015b*; *Gostic et al., 2016*; *Cobey and Hensley, 2017*), and variation in exposure history is a plausible cause of the person-to-person variation we observe in the antigenic effects of viral mutations. A role for exposure history is further suggested by the lack of variation in immune selection among ferrets that have been experimentally infected just once with a defined viral strain. However, our results do not formally distinguish whether the differences between human and ferret sera are due to exposure history or species-to-species variation in immunodominance hierarchies (*Liu et al., 2018*; *Angeletti et al., 2017*; *Fonville et al., 2016*; *Altman et al., 2015*)—rigorously addressing this question will require mapping the immune selection exerted by children that have been infected with influenza virus just once. In either case, the differences between human and ferret sera underscore the shortcomings of using singly infected ferrets as the primary tool for characterizing viral antigenicity, as has previously been shown for H1N1 (*Li et al., 2013*).

Overall, our work uses a new approach to show that polyclonal human sera can strongly select single antigenic mutations in HA, and that the effects of these mutations vary greatly person to person. Expanding this approach should shed further light on the causes of this variable focusing, and its implications for viral evolution and disease susceptibility.

## Materials and methods

### Data and computer code

Deep sequencing data are on the Sequence Read Archive under BioSample accession numbers SAMN10183083 and SAMN11310465 (viral libraries selected with monoclonal antibodies), SAMN10183146 (monoclonal antibody selection controls), SAMN11310372 (viral libraries selected with human sera), SAMN11310371 (viral libraries selected with ferret sera), SAMN11341221 (viral libraries selected with human sera spiked with monoclonal antibody), and SAMN11310373 (serum selection controls).

The computer code used to analyze the data and generate the paper figures is available on GitHub (*Lee and Bloom, 2019*; copy archived at https://github.com/elifesciences-publications/map_flu_serum_Perth2009_H3_HA). Key processed data are tracked in this GitHub repository as detailed in the repository's README file, and are also available in the Supplementary files for this manuscript.

### HA numbering

All sites are referred to in H3 numbering unless indicated otherwise. The signal peptide is in negative numbers, the HA1 subunit in plain numbers, and the HA2 subunit in numbers denoted with '(HA2).' Sequential 1, 2, ... numbering of the Perth/2009 H3 HA can be converted to H3 numbering simply by subtracting 16 for the H1 subunit, and subtracting 345 for the HA2 subunit. A file mapping

sequential 1, 2, . . . numbering to H3 numbering is at https://github.com/jbloomlab/map_flu_serum_Perth2009_H3_HA/blob/master/data/H3renumbering_scheme.csv.

## Perth/2009 HA virus used in this study

The 'wild-type' Perth/2009 HA used in this study is that from *Lee et al. (2018)*. This HA has two amino-acid mutations (G78D and T212I) relative to the most common Perth/2009 HA sequences in Genbank (there are several variants in Genbank). These two mutations improve viral growth in cell culture (*Lee et al., 2018*). The sequence of HA with these mutations is provided in *Lee et al. (2018)* and at https://github.com/jbloomlab/map_flu_serum_Perth2009_H3_HA/blob/master/data/Perth09_HA_reference.fa. Unless otherwise indicated, when this paper refers to the 'Perth/2009 HA' it means this HA. As described below, all other genes in this virus were derived from A/WSN/1933, a H1N1 strain that grows efficiently in cell culture and was heavily adapted from its original human isolate predecessor by passaging in mouse brain in the lab (*Stuart-Harris, 1939*; *Sun et al., 2010*).

## Monoclonal antibodies

The monoclonal antibodies in *Figure 2* are described in *Zost et al. (2019)*, *Zost et al. (2017)* and *Henry Dunand et al. (2015)*. These antibodies were isolated from peripheral blood mononuclear cells of human donors 7 days post-vaccination with the 2010–2011 influenza vaccine containing the A/Victoria/210/2009 strain as the H3N2 component, using the approach in *Smith et al. (2009)*. The $V_H$ and $V_L$ chains were amplified using single-cell RT-PCR, and cloned into human IgG expression vectors. To produce the monoclonal antibodies, 293 T cells were transfected with plasmids encoding the heavy and light chains, and the antibodies were purified using protein A/G affinity purification.

## Human sera

The human sera in *Figure 3* came from the Infectious Disease Sciences Biospecimen Repository at the Vaccine and Infectious Disease Division (VIDD) of the Fred Hutchinson Cancer Research Center. These sera were collected in Seattle, WA, from healthy prospective bone marrow transplant donors, who provided written consent for the use of their sera in research, and the sera were banked at –20° C. We used neutralization assays to screen sera collected from 16 individuals from 2008 and 2010, and selected the four that neutralized virus with the Perth/2009 HA at $\geq$ 1:40 dilution. No information was available on the influenza vaccination or infection status of these individuals.

The human sera in *Figure 5* were collected at the Wistar Institute, Philadelphia, PA, from four individuals pre-vaccination and four weeks post-vaccination with the 2015–2016 Northern Hemisphere influenza vaccine. The H3N2 component of this vaccine was A/Switzerland/9715293/2013, which is an antigenic successor of Perth/2009 by several vaccine updates: Perth/2009 was the Northern Hemisphere vaccine strain in 2010–2012, A/Victoria/361/2011 was the strain in 2012–2014, A/Texas/50/2012 was the strain in 2014–2015, and Switzerland/2013 was the strain in 2015–2016. The Wistar Institute IRB approved the study collecting these samples.

The dates of collection of the human sera are rounded to the nearest year in this paper.

## Ferret sera

The serum samples denoted 'ferret-Pitt-#' were collected from three ferrets pre-infection and again 23 days post-infection at the University of Pittsburgh. Each ferret was infected with Perth/2009 (H3N2) virus that carries the exact same HA sequence used for the studies in this paper. Ferret studies were performed under an approved IACUC protocol (#16077170) at the University of Pittsburgh, an AAALAC accredited facility.

The serum samples denoted 'ferret-WHO*' were collected from two ferrets post-infection at the World Health Organization Collaborating Centre in Melbourne, Australia. One ferret was infected with the Collaborating Centre's version of the Perth/2009 H3N2 virus, and the second ferret was infected with the Collaborating Centre's version of the A/Victoria/361/2011 H3N2 strain. The sera were collected in bleeds performed 14 days after intranasal infection with virus. Experiments were conducted with approval from the University of Melbourne Biochemistry and Molecular Biology, Dental Science, Medicine, Microbiology and Immunology, and Surgery Animal Ethics Committee, in accordance with the NHMRC Australian code of practice for the care and use of animals for scientific purposes.

## Serum preparation

All sera were treated with receptor-destroying enzyme (RDE) and heat-inactivated before use in our experiments, using a protocol adapted from *Zost et al. (2017)*. The RDE treatment was to ensure that the viral libraries would not bind to residual sialic acids present in the serum. One vial of lyophilized RDE II (Seiken, Cat No. 370013) was first resuspended in 20 mL PBS. We then incubated 100 µL of serum with 300 µL of RDE solution at 37°C for 2.5 hr. We next heat-inactivated the serum and RDE by incubating at 55°C for 30 min. Finally, we centrifuged the serum at 20,000 x *g* for 20 min to pellet any precipitated material, collected the supernatant, aliquoted, and stored at –80°C.

## Neutralization assays

Neutralization assays were performed using influenza viruses carrying GFP in the PB1 segment. The 'wild-type' Perth/2009 viruses were generated using reverse genetics with the following plasmids: pHW-Perth2009-HA-G78D-T212I (*Lee et al., 2018*), pHH-PB1flank-eGFP (which encodes GFP on the PB1 segment; *Bloom et al., 2010*), and pHW181-PB1, pHW183-PA, pHW185-NP, pHW186-NA, pHW187-M, and pHW188-NS (which encode genes from A/WSN/1933 for the other six segments; *Hoffmann et al., 2000*). The individual amino-acid mutant viruses tested in the neutralization assays were generated from the same plasmids except the indicated mutation was introduced into the pHW-Perth2009-HA-G78D-T212I plasmid. The 'syn' mutant shown in *Figure 3* has a synonymous F193F mutation.

To generate the viruses, we transfected a co-culture of $4 \times 10^5$ 293T-CMV-PB1 (*Bloom et al., 2010*) and $0.5 \times 10^5$ MDCK-SIAT1-CMV-PB1-TMPRSS2 (*Lee et al., 2018*) cells with the eight reverse genetics plasmids and the pHAGE2-EF1aInt-TMPRSS2-IRES-mCherry-W plasmid (*Lee et al., 2018*). The transfections were performed in D10 media (DMEM, supplemented with 10% heat-inactivated FBS, 2 mM L-glutamine, 100 U of penicillin per milliliter, and 100 µg of streptomycin per milliliter). Each well received a transfection mixture of 100 µL DMEM, 3 µL BioT transfection reagent, and 250 ng of each plasmid. We changed the media in each well to 2 mL IGM (Influenza Growth Media, consisting of Opti-MEM supplemented with 0.01% heat-inactivated FBS, 0.3% BSA, 100 U of penicillin per milliliter, 100 µg of stremptomycin per milliliter, and 100 µg of calcium chloride per milliliter) eight hours post-transfection. At approximately 56 hr post-transfection, transfection supernatants were harvested, clarified by centrifugation at 2000 x*g* for 5 min, aliquoted, and frozen at −80°C. To titer the GFP-carrying viruses, we plated $1 \times 10^5$ MDCK-SIAT1-CMV-PB1-TMPRSS2 cells per well in 12-well plates in IGM. We infected cells with dilutions of viral supernatant four hours after plating. At 16 hr post-infection, we chose wells that appeared to show 1% to 10% of cells that were GFP-positive, and determined the exact fraction of GFP-positive cells by flow cytometry to calculate the titer of infectious particles per µl.

The neutralization assays were performed using these GFP-expressing viruses as described previously (*Hooper and Bloom, 2013*; *Doud et al., 2018*; see also *Bloom and Lee, 2019*; copy archived at https://github.com/elifesciences-publications/flu_PB1flank-GFP_neut_assay). All neutralization curves represent the mean and standard error of three replicate curves run on the same 96-well plate. The curve fits are Hill-like curves with the bottom and top constrained to zero and one, and were fit using the neutcurve Python package (https://jbloomlab.github.io/neutcurve/), version 0.3.0. Detailed information on the curve fitting is available on the project GitHub repository at https://github.com/jbloomlab/map_flu_serum_Perth2009_H3_HA/blob/master/results/notebooks/analyze_neut.md. The curve-fit parameters (e.g., IC50s) for all of the neutralization curves shown in this paper are in *Supplementary file 2*. The dashed vertical lines in *Figure 3B*, *Figure 4B*, *Figure 5*, and *Figure 6* indicate the average concentration of serum used across the replicates.

## Mutant virus libraries

For mutational antigenic profiling of some replicates of antibodies 4C01 and 1C04, we used the exact same mutant virus libraries described in *Lee et al. (2018)*. Specifically, we used those mutant virus libraries for the replicate labeled as 'lib2' for 4C01 and all replicates of 1C04 in *Figure 2—figure supplement 2*.

However, the titers of the viral libraries from *Lee et al. (2018)* were not sufficient for all the mutational antigenic profiling, so for most of the experiments in the current paper we generated new triplicate stocks of mutant virus libraries using influenza reverse-genetics (*Hoffmann et al., 2000*).

This required re-generating the mutant plasmids and mutant virus libraries as described below. These libraries have similar properties to those described in *Lee et al. (2018)*, but since they are not exactly the same we re-validated them by deep sequencing the mutant plasmid and mutant virus libraries. The full analysis of this deep sequencing along with plots showing all relevant statistics about the libraries is in *Supplementary file 5*.

To create the new mutant plasmid libraries, we used two rounds of codon mutagenesis (*Bloom, 2014*; *Dingens et al., 2017*) to introduce all of the possible codon mutations into the Perth/2009 HA in the pHH21 backbone (*Neumann et al., 1999*). The plasmid libraries were generated independently in biological triplicate, starting from independent preps of the wildtype plasmid, using the protocol described in *Lee et al. (2018)*. The mutant amplicons were then cloned at high efficiency into the pHH21 vector using digestion with BsmBI, ligation with T4 DNA ligase, and electroporation into ElectroMAX DH10B competent cells (Invitrogen 18290015). We obtained >4 million transformants for each replicate library. We then scraped the plates, expanded the cultures in liquid LB + ampicillin at 37°C for 3 hr with shaking, and maxiprepped the cultures. Sanger sequencing of 29 randomly chosen clones showed an average mutation rate of 1.6 codon mutations per clone (*Supplementary file 5*).

To generate the mutant virus libraries by reverse genetics, we transfected 40 wells of six-well plates for each library. Each well contained a co-culture of $5 \times 10^5$ 293T-CMV-PB1 and $0.5 \times 10^5$ MDCK-SIAT1-TMPRSS2 cells in D10 media. We transfected with 250 ng each of pHH-mutant-HA library (or wildtype control), the pHAGE2-EF1aInt-TCmut-P09-HA Perth/2009 HA protein expression plasmid (which expresses the HA protein from wild type Perth/2009 HA), the pHW18* series of plasmids (*Hoffmann et al., 2000*) for all non-HA viral genes, and pHAGE2-EF1aInt-TMPRSS2-IRES-mCherry-W. The sequence of the pHAGE2-EF1aInt-TCmut-P09-HA protein expression plasmid is in *Supplementary file 6*. We changed the media in each well to 2 mL IGM 8 hr post-transfection. At 45 hr post-transfection, the transfection supernatants were harvested, clarified by centrifugation at 2,000 x*g* for 5 min, aliquoted, frozen at −80°C, and titered in MDCK-SIAT1-TMPRSS2 cells. The titers were 2543, 3162, 1000, and 4739 $TCID_{50}$ per microliter for the three library replicates and the wild-type control, respectively.

We passaged 1.125 x $10^6$ $TCID_{50}$ of the transfection supernatants for each library at an MOI of 0.005 $TCID_{50}$ per cell. We did this by plating 3 x $10^6$ MDCK-SIAT1-TMPRSS2 cells per layer of five 5-layered 875 $cm^2$ flasks (Corning, 353144) in D10 media, and allowed the cells to grow for 24 hr, at which time they were at ~9 x $10^6$ cells per layer. We then removed the D10 media from each flask, washed with 50 mL PBS and replaced the media with 130 mL per flask of an inoculum of 1.73 $TCID_{50}$ of virus per microliter in IGM. At 3 hr post-infection, we replaced the inoculum with fresh IGM for each replicate. We then collected virus supernatant at 42 hr post-infection, clarified the supernatant by centrifuging at 2000 x*g* for 5 min, aliquoted, froze at −80°C, and titered in MDCK-SIAT1-TMPRSS2 cells. The titers were 1000, 14677, and 6812 $TCID_{50}$ per microliter for the three library replicates, respectively. The mutant plasmids and mutant viruses were then deep sequenced as in *Lee et al. (2018)* to demonstrate that there was good coverage of mutations in both as described in *Supplementary file 5*. These libraries were used for all the mutational antigenic profiling except for the subset of antibody replicates mentioned at the beginning of this subsection.

As described above, all viral propagation used MDCK-SIAT1-CMV-TMPRSS2 cells. These cells were tested for Mycoplasma at the Fred Hutch Research Cell Bank core, and confirmed to be Mycoplasma negative.

## Mutational antigenic profiling

We performed the mutational antigenic profiling using the basic process described in *Doud et al. (2017)*. For each serum and for the serum-antibody spike-in experiments, we performed three biological replicates of mutational antigenic profiling each using an independently generated mutant virus library. For each antibody, we performed either two or three biological replicates each using an independently generated mutant virus library as indicated in *Figure 2—figure supplement 2*. The reason that we only performed two biological replicates for some antibodies is that the noise is less for antibodies than sera.

For the mutational antigenic profiling, we diluted each virus library to $10_6$ $TCID_{50}$ per mL, and incubated the virus dilution with an equal volume of antibody and/or serum at the intended dilution at 37°C for 1.5 hr. The dilutions for all samples are in *Supplementary file 5*, and account for the

initial 1:4 dilution of serum during the RDE treatment. These dilutions were generally chosen with the goal of having 1% to 10% of the virus library survive the serum or antibody treatment. We infected between $2 \times 10^5$ and $4 \times 10^5$ MDCK-SIAT1-TMPRSS2 cells with virus-antibody or virus-serum mix. At 15 hr post-infection, we extracted RNA from the cells, and then reverse-transcribed and PCR amplified full-length HA as in *Lee et al. (2018)*. We then deep sequenced these HAs using a barcoded-subamplicon sequencing strategy to ensure high accuracy. This general sequencing approach was first applied to viral deep mutational scanning by *Wu et al. (2014)*. The exact approach we used is described (*Doud and Bloom (2016)*, with the primers for Perth/2009 given in *Lee et al. (2018)*; a more general description of the approach is at https://jbloomlab.github.io/dms_tools2/bcsubamp.html. The sequencing was performed using 2 x 250 nucleotide paired-end reads on Illumina HiSeq 2500's at the Fred Hutchinson Cancer Research Center Genomics Core.

To estimate the overall fraction of virions in the library surviving immune selection, we used qRT-PCR against the viral NP and GAPDH, as described in *Doud et al. (2017)*. Briefly, we made duplicate 10-fold serial dilutions of each virus library to create a standard curve of infectivity. We then performed qPCR for the standard curve of infectivity as well as each library-selected sample. A linear regression line relating the logarithm of the viral infectious dose in the standard curve to the difference in Ct values between NP and GAPDH was used to interpolate the fraction surviving for each selection. The measured percent surviving for each library are in *Supplementary file 4*, and are also plotted in figure supplements for each figure showing mutational antigenic profiling results.

## Analysis of deep sequencing data and visualization of results

The deep sequencing data were analyzed using dms_tools2 (*Bloom, 2015*) version 2.4.16, which is available at https://jbloomlab.github.io/dms_tools2/. Briefly, we first determined the counts of each codon at each site in both the immune-selected and mock-selected samples. These counts are at https://github.com/jbloomlab/map_flu_serum_Perth2009_H3_HA/tree/master/results/renumbered_codoncounts. These counts were then processed to compute the *differential selection* on each amino-acid mutation at each site, which is our measure of immune selection. The differential selection statistic is described in *Doud et al. (2017)* (see also https://jbloomlab.github.io/dms_tools2/diffsel.html), and represents the log enrichment of each mutation relative to wildtype in the immune-selected sample versusa mock-selected control. The differential selection values for each replicate are at https://github.com/jbloomlab/map_flu_serum_Perth2009_H3_HA/tree/master/results/diffsel.

To visualize the differential selection, we took the median across replicates of the differential selection for each mutation at each site—these median values are displayed in all logo plots. The numerical values of these across-replicate median differential selection values are in *Supplementary file 7*. The figures in this paper visualize the differential selection in two ways. First, the line plots show the total positive differential selection at each site. Second, the logo plots show the differential selection for each positively selected amino acid at key sites. Note that in both cases, negative differential selection is *not* shown. These line and logo plots were created using the dmslogo software package (version 0.2.3), which is available at https://jbloomlab.github.io/dmslogo/.

We chose which sites to show in the logo plots in the figures by identifying strong or 'significant' sites of immune selection for each serum using the approach described in *Dingens et al. (2019)* (excluding the pre-vaccination or pre-infection samples). Each figure panel then shows all sites that were 'significant' for any sera or antibody in that panel. This 'significance' calculation is heuristic, and involves using robust regression to fit a gamma distribution to all of the positive site differential selection values, equating the p-value to the fraction of the distribution $\geq$ that site's differential selection, and then calling 'significant' sites that have a false discovery rate $\leq 0.05$. The code that performs this analysis is at https://jbloomlab.github.io/dms_tools2/dms_tools2.plot.html#dms_tools2.plot.findSigSel. In addition, logo plots for all sites for the across-replicate medians for each serum/antibody are in *Supplementary file 8*.

A detailed notebook with the code for all of the foregoing analyses along with explanations and many additional plots is at https://github.com/jbloomlab/map_flu_serum_Perth2009_H3_HA/blob/master/results/notebooks/analyze_map.md.

## Beta diversity

To quantify the variation among sites of selection for different groups of sera (i.e., the ferret sera, the 2009–2010 human sera, and the 2015 post-vaccination human sera), we used the beta diversity statistic from ecology. For each serum in a group, we calculated the fraction of the total positive site differential selection attributable to each site. We then computed the beta diversity of this selection fraction among sites, using the Simpson index to quantify diversity, following the method of *Jost (2007)*. Specifically, let $p_{r,s}$ be the total fraction of all positive site differential selection for serum $s$ that is attributable to site $r$ (so $1 = \sum_r p_{r,s}$). The Simpson concentration index for serum $s$ is then $\lambda_s = \sum_r (p_{r,s})^2$. For a group of sera, the gamma diversity index (total diversity across all sera in the group) is $\lambda^\gamma = \sum_r \langle p_r \rangle^2$ where $\langle p_r \rangle$ is the average of $p_{r,s}$ across all sera $s$. The corresponding true gamma diversity is $D_\gamma = 1/\lambda^\gamma$. Likewise, the alpha diversity index is simply the mean of the index for each serum, $\lambda^\alpha = \langle \lambda_s \rangle$, and the true alpha diversity is $D_\alpha = 1/\lambda^\alpha$. The beta diversity is then simply $D_\beta = D_\gamma/D_\alpha$.

Code that implements this calculation has been added to dms_tools2 (*Bloom, 2015*) version 2.4.16 (see https://jbloomlab.github.io/dms_tools2/dms_tools2.diffsel.html#dms_tools2.diffsel.beta_diversity).

## Protein structures

The protein structures are all PDB 4O5N, which is the structure of HA from the A/Victoria/361/2011 (H3N2) strain (*Lee et al., 2014*). The residues are colored by the positive site differential selection values. The visualizations were generated using nglview (*Nguyen et al., 2018*) via the Python wrapper package dms_struct (https://jbloomlab.github.io/dms_struct/). Interactive mybinder instances of the notebooks that can be used to rotate and zoom in on the structures are available at the following weblinks:

- Antibodies targeting antigenic region B (*Figure 2*): https://mybinder.org/v2/gh/jbloomlab/map_flu_serum_Perth2009_H3_HA/master?urlpath=%2Fapps%2Fresults%2Fnotebooks%2Fmap_on_struct_antibody_region_B.ipynb
- Antibodies target lower on HA's head (*Figure 2*): https://mybinder.org/v2/gh/jbloomlab/map_flu_serum_Perth2009_H3_HA/master?urlpath=%2Fapps%2Fresults%2Fnotebooks%2Fmap_on_struct_antibody_lower_head.ipynb
- The 'VIDD' human sera (*Figure 3*): https://mybinder.org/v2/gh/jbloomlab/map_flu_serum_Perth2009_H3_HA/master?urlpath=%2Fapps%2Fresults%2Fnotebooks%2Fmap_on_struct_VIDD_sera.ipynb
- The 'Hensley' human sera (*Figure 5*): https://mybinder.org/v2/gh/jbloomlab/map_flu_serum_Perth2009_H3_HA/master?urlpath=%2Fapps%2Fresults%2Fnotebooks%2Fmap_on_struct_Hensley_sera.ipynb
- The ferret sera (*Figure 6*): https://mybinder.org/v2/gh/jbloomlab/map_flu_serum_Perth2009_H3_HA/master?urlpath=%2Fapps%2Fresults%2Fnotebooks%2Fmap_on_struct_ferret.ipynb

## Mutation frequencies in natural sequences

For *Figure 8*, we first identified sites that were under strong or 'significant' selection from any of the human serum samples (excluding the pre-vaccination samples in *Figure 5*) using the approach described above. There were 16 such sites; these are the ones shown in the logo plots in *Figure 3A* or *Figure 5A*. For each such site, we then examined the frequency of different amino-acid identities from 2007 to 2019 (see https://github.com/jbloomlab/map_flu_serum_Perth2009_H3_HA/blob/master/results/notebooks/analyze_natseqs.md). This analysis identified nine sites where a new amino-acid identity reached at least 5% frequency. For these nine sites, we then took images of the amino-acid frequencies over time from the Nextstrain website (https://nextstrain.org/) (*Hadfield et al., 2018*; *Neher and Bedford, 2015*) and used them to create *Figure 8*.

## Acknowledgements

We thank Hai Nguyen for assistance with using nglview. We thank Kanta Subbarao for helpful advice. We thank Andrea Loes, Allison Greaney, Tal Einav, and John Huddleston for helpful comments on

the manuscript. We thank the Fred Hutchinson Cancer Research Center Genomics Core for performing the Illumina deep sequencing. The authors declare no competing interests.

## Additional information

### Funding

| Funder | Grant reference number | Author |
| --- | --- | --- |
| National Institutes of Health | R01 AI127893 | Trevor Bedford<br>Jesse D Bloom |
| National Institutes of Health | R01 AI141707 | Jesse D Bloom |
| National Institutes of Health | R01 AI113047 | Scott E Hensley |
| National Institutes of Health | R01 AI108686 | Scott E Hensley |
| National Institutes of Health | HHSN272201400005C | Scott E Hensley<br>Jesse D Bloom |
| Burroughs Wellcome Fund | | Scott E Hensley<br>Jesse D Bloom |
| National Institutes of Health | F30 AI136326 | Juhye M Lee |
| National Institutes of Health | U54GM111274 | Juhye M Lee |
| Pew Charitable Trusts | | Trevor Bedford |
| National Institutes of Health | R35 GM119774 | Trevor Bedford |
| National Institutes of Health | U19 AI117891 | Trevor Bedford |
| Howard Hughes Medical Institute | | Jesse D Bloom |

The funders had no role in study design, data collection and interpretation, or the decision to submit the work for publication.

### Author contributions

Juhye M Lee, Conceptualization, Formal analysis, Investigation, Methodology, Writing—original draft, Writing—review and editing; Rachel Eguia, Investigation, Writing—review and editing; Seth J Zost, Conceptualization, Resources, Investigation, Writing—review and editing; Saket Choudhary, Software; Patrick C Wilson, Terry Stevens-Ayers, Michael Boeckh, Aeron C Hurt, Seema S Lakdawala, Resources, Writing—review and editing; Trevor Bedford, Software, Funding acquisition, Visualization, Writing—review and editing; Scott E Hensley, Conceptualization, Resources, Writing—review and editing; Jesse D Bloom, Conceptualization, Software, Formal analysis, Supervision, Funding acquisition, Writing—original draft, Writing—review and editing

### Author ORCIDs

Seth J Zost (iD) http://orcid.org/0000-0001-6712-5076
Saket Choudhary (iD) http://orcid.org/0000-0001-5202-7633
Trevor Bedford (iD) http://orcid.org/0000-0002-4039-5794
Seema S Lakdawala (iD) http://orcid.org/0000-0002-7679-2150
Jesse D Bloom (iD) https://orcid.org/0000-0003-1267-3408

### Ethics

Human subjects: This study used de-identified human sera collected from healthy volunteers at the Fred Hutchinson Cancer Research Center and the Wistar Institute. The respective Institutional Review Boards approved collection and use of these sera.
Animal experimentation: This study used sera from ferrets experimentally infected with influenza virus at the University of Pittsburgh and the World Health Organization Collaborating Centre in Melbourne. The experiments at the University of Pittsburgh were performed under an approved IACUC protocol (#16077170) at an AAALAC accredited facility. The experiments in Melbourne were

conducted with approval from the University of Melbourne Biochemistry & Molecular Biology, Dental Science, Medicine, Microbiology & Immunology, and Surgery Animal Ethics Committee, in accordance with the NHMRC Australian code of practice for the care and use of animals for scientific purposes.

## Decision letter and Author response

Decision letter https://doi.org/10.7554/eLife.49324.048
Author response https://doi.org/10.7554/eLife.49324.049

## Additional files

### Supplementary files

• Supplementary file 1. Classically defined antigenic regions of H3N2 HA and their relationship to sites of strong selection in our mapping experiments. The first column of this Excel document lists all sites of relevance in H3 numbering. The second and third column indicates which sites have been assigned to antigenic regions A, B, C, D, or E according to Table 1 of *Wiley et al. (1981)* or SI Table 1 of *Shih et al. (2007)* (sites listed in those papers but not assigned a labeled antigenic region are not included in the columns). The remaining columns indicate which sites are under strong selection from any antibody/serum in each set.
DOI: https://doi.org/10.7554/eLife.49324.023

• Supplementary file 2. The curve fit parameters for all neutralization curves shown in the figures. The IC50 values are *not* extrapolated, and so are shown as upper or lower bounds if they fall outside the range of the measurements. For sera, the IC50s are the serum dilution; for antibodies they are the antibody concentration in µg/ml. This CSV file is also available at https://github.com/jbloomlab/map_flu_serum_Perth2009_H3_HA/blob/master/results/neutralization_assays/neut_assay_figs_fit_params.csv.
DOI: https://doi.org/10.7554/eLife.49324.024

• Supplementary file 3. The serum dilution or antibody concentration used for each replicate of the mutational antigenic profiling. For sera, the values indicate the dilution of serum. For antibodies, they are the concentration in µµg/ml. For serum/antibody mixes, they are the dilution of serum followed by the antibody concentration in µg/ml. These dilutions/concentrations were chosen to give the desired percent of viral infectivity remaining for the libraries after treatment (see *Supplementary file 4*). The dashed vertical lines in *Figure 3B*, *Figure 4B*, *Figure 5*, and *Figure 6* indicate the average concentration of serum used across the replicates. This CSV file is also available at https://github.com/jbloomlab/map_flu_serum_Perth2009_H3_HA/blob/master/results/selection_tables/serum_dilution_table.csv.
DOI: https://doi.org/10.7554/eLife.49324.025

• Supplementary file 4. The percent of the overall viral library that retained infectivity after incubation with serum or antibody. This CSV file is also at https://github.com/jbloomlab/map_flu_serum_Perth2009_H3_HA/blob/master/results/selection_tables/serum_dilution_table.csv.
DOI: https://doi.org/10.7554/eLife.49324.026

• Supplementary file 5. HTML rendering of Jupyter notebook that analyzes the mutant virus libraries generated by reverse genetics.
DOI: https://doi.org/10.7554/eLife.49324.027

• Supplementary file 6. A GenBank file providing the full sequence of the protein expression plasmid pHAGE2-EF1aInt-TCmut-P09-HA, which encodes for the wildtype Perth/2009 HA sequence.
DOI: https://doi.org/10.7554/eLife.49324.028

• Supplementary file 7. Numerical values of the differential selection (immune selection) values for each amino-acid at each site after taking the median across replicates. These are the values plotted in the line and logo plots in the main figures. This tidy-format CSV file is also available at https://github.com/jbloomlab/map_flu_serum_Perth2009_H3_HA/blob/master/results/avgdiffsel/avg_sel_tidy.csv.
DOI: https://doi.org/10.7554/eLife.49324.029

• Supplementary file 8. Logo plots of the positive differential selection for all sites in HA for each serum and antibody selection. The main figures in this paper just zoom in on the key sites of selection. These PDFs are also available at https://github.com/jbloomlab/map_flu_serum_Perth2009_H3_HA/tree/master/results/avgdiffsel/full_logo_plots.
DOI: https://doi.org/10.7554/eLife.49324.030

• Supplementary file 9. Key resources table listing the most crucial reagents and computer software used in the study.
DOI: https://doi.org/10.7554/eLife.49324.031

• Transparent reporting form
DOI: https://doi.org/10.7554/eLife.49324.032

## Data availability

All computer code and processed data are on GitHub at https://github.com/jbloomlab/map_flu_serum_Perth2009_H3_HA (copy archived at https://github.com/elifesciences-publications/map_flu_serum_Perth2009_H3_HA). Deep sequencing data are on the Sequence Read Archive under BioSample accession numbers SAMN10183083 and SAMN11310465 (viral libraries selected with monoclonal antibodies), SAMN10183146 (monoclonal antibody selection controls), SAMN11310372 (viral libraries selected with human sera), SAMN11310371 (viral libraries selected with ferret sera), SAMN11341221 (viral libraries selected with human sera spiked with monoclonal antibody), and SAMN11310373 (serum selection controls).

The following datasets were generated:

| Author(s) | Year | Dataset title | Dataset URL | Database and Identifier |
|---|---|---|---|---|
| Lee JM, Eguia R, Zost SJ, Choudhyar S, Wilson PC, Bedford T, Stevens-Ayers T, Bo eckh M, Hurt A, Lakdawala SS, Hensley SE, Bloom JD | 2019 | Mutational antigenic profiling of antibodies | https://www.ncbi.nlm.nih.gov/biosample/SAMN10183083 | Sequence Read Archive, SAMN10183083 |
| Lee JM, Eguia R, Zost SJ, Choudhyar S, Wilson PC, Bedford T, Stevens-Ayers T, Bo eckh M, Hurt A, Lakdawala SS, Hensley SE, Bloom JD | 2019 | Mutational antigenic profiling of antibodies | https://www.ncbi.nlm.nih.gov/biosample/SAMN11310465 | Sequence Read Archive, SAMN11310465 |
| Lee JM, Eguia R, Zost SJ, Choudhyar S, Wilson PC, Bedford T, Stevens-Ayers T, Bo eckh M, Hurt A, Lakdawala SS, Hensley SE, Bloom JD | 2019 | Mutational antigenic profiling antibody controls | https://www.ncbi.nlm.nih.gov/biosample/SAMN10183146 | Sequence Read Archive, SAMN10183146 |
| Lee JM, Eguia R, Zost SJ, Choudhyar S, Wilson PC, Bedford T, Stevens-Ayers T, Bo eckh M, Hurt A, Lakdawala SS, Hensley SE, Bloom JD | 2019 | Mutational antigenic profiling of human sera | https://www.ncbi.nlm.nih.gov/biosample/SAMN11310372 | Sequence Read Archive, SAMN11310372 |
| Lee JM, Eguia R, Zost SJ, Choudhyar S, Wilson PC, Bedford T, Stevens-Ayers T, Bo eckh M, Hurt A, Lakdawala SS, Hensley SE, Bloom JD | 2019 | Mutational antigenic profiling of ferret sera | https://www.ncbi.nlm.nih.gov/biosample/SAMN11310371 | Sequence Read Archive, SAMN11310371 |

| | | | | |
|---|---|---|---|---|
| Lee JM, Eguia R, Zost SJ, Choudhyar S, Wilson PC, Bedford T, Stevens-Ayers T, Bo eckh M, Hurt A, Lakdawala SS, Hensley SE, Bloom JD | 2019 | Mutational antigenic profiling of antibody spike in | https://www.ncbi.nlm.nih.gov/biosample/SAMN11341221 | Sequence Read Archive, SAMN11341221 |
| Lee JM, Eguia R, Zost SJ, Choudhyar S, Wilson PC, Bedford T, Stevens-Ayers T, Bo eckh M, Hurt A, Lakdawala SS, Hensley SE, Bloom JD | 2019 | Mutational antigenic profiling sera controls | https://www.ncbi.nlm.nih.gov/biosample/SAMN11310373 | Sequence Read Archive, SAMN11310 373 |

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
