## [Decision Letter]

Thank you for submitting your article "Mapping person-to-person variation in viral mutations that escape polyclonal serum targeting influenza hemagglutinin" for consideration by *eLife*. Your article has been reviewed by three peer reviewers, including Marc Lipsitch as the Reviewing Editor and Reviewer #3, and the evaluation has been overseen by a Reviewing Editor and Karla Kirkegaard as the Senior Editor.

The reviewers have discussed the reviews with one another and the Editors have written this decision to help you prepare a revised submission.

Summary:

This is a well-conceived, executed, and written paper. It uses a novel (though previously described) technique developed in the Bloom lab to define the routes of antigenic escape from monoclonal antibodies and polyclonal sera in influenza HA. It represents a major step forward in our understanding, in particular:

1) Demonstrating that, surprisingly, significant escape can be obtained by a single AA change in HA despite multiple specificities of the response.

2) Different individuals at a given point in time (approximately) have very different polyclonal specificities and thus routes of escape.

3) The situation is different in ferrets that have received only one flu infection, suggesting (though not demonstrating, as this would require first-infection humans and multiple-infection ferrets) that the difference might be history of infection.

Obviously this raises many questions, most notably that raised in the Discussion of why a single mutation can create significant escape. But this is a large and comprehensive study that suggests a continuing line of research, and is certainly substantial enough for an *eLife* paper. Congratulations to the authors. Also commendable are the depth of assay validation shown in the figures, the availability of code, and the explanation of the numbering system used in the manuscript, which is easy to understand and makes the data much more accessible for non-aficionados.

Essential revisions:

1) The authors should try to define a more quantitative null expectation of person-to-person variation. While the results show obvious differences in the escape mutants selected between individuals, is there a way to quantify exactly how the diversity of escape mutants among humans differs from the diversity of escape mutants among ferrets? Perhaps a beta diversity-like metric would provide a more precise measure of the difference in mutational profiles between individuals and facilitate a quantitative comparison of the human and ferret results.

2) While the results linking the natural diversity of influenza to the selected mutants are suggestive, the reviewers found claims about the relationship to global evolution difficult to fully justify. Specifically, given the small number of individuals in this study, it is unclear how we can classify a particular selective pressure being present in "only a subset" vs "many individuals." If such claims are to be made, a more explicit link needs to be made between the results presented in the manuscript the fraction of the global population that may exert a particular selective pressure on influenza. Also, while the results involving the 53 year-old patient suggest stability of selective pressure over short timescales, that stability over longer timescales may be important to explaining global patterns of influenza evolution. This additional work is not essential, but tempering this section may be called for.

3) The authors focus only on in vitro neutralization, which is fine. However, antibodies have many functions and in vitro neutralization doesn't always correlate well with protection in vivo. Other antibodies that are present (HA side binders, stalk binders, non-neutralizing antibodies, antibodies that bind HA but inhibit NA etc.) might not show much activity in an in vitro entry inhibition assay but might be highly relevant in vivo. There is nothing wrong with using an in vitro assay here and the take away points from the study are important. But the above needs to be discussed and it needs to be pointed out that many other factors might influence drift. Also please discuss why such high titers were present (>1:10,000), this will be a surprise to many readers.

4) The authors dilute the serum to approximately 1 IC90. By doing that, the effect of other antibodies present at lower titers that potentially target other sites is diminished or completely eliminated. The serum dilutions used should be stated and this should be discussed.

5) While IgG (which forms the majority of serum antibodies) protects the lung and is important for lower respiratory tract infections it is unlikely that it drives drift. Please add to the Discussion that the vast majority of influenza virus infections are restricted to the upper respiratory tract which is protected by IgA1.

6) In a way it is concerning that mAb spiked into serum kind of 'takes over', diminishing the effect of the antibodies in serum in the set up used. This might be an artifact of the library system or the fact that the serum is strongly diluted. But this discrepancy needs to be pointed out because it suggests that one dominant antibody clonotype in serum could skew what the authors see when they perform their analysis. It really begs the question why serum neutralizing activity seems to become irrelevant as soon as highly neutralizing mAb is present? One possible explanation is, that the mAb actually sterically blocks binding of the dominant neutralizing antibody in serum? But then, mutants that escape the mAb should still be neutralized by serum antibody? Please clarify.

7) Discussion section: "the importance of history is demonstrated by the lack of variation among ferrets that have been experimentally infected just once with a defined viral strain." This is exactly right, but then there should not be a claim anywhere in the paper that humans and ferrets are different, but rather that adult humans (with presumed multiple infections) and ferrets after a single infection are different. I agree that history rather than host biology seems the more likely explanation, but in the current study the two are confounded.

---

## [Author Response]

Essential revisions:1) The authors should try to define a more quantitative null expectation of person-to-person variation. While the results show obvious differences in the escape mutants selected between individuals, is there a way to quantify exactly how the diversity of escape mutants among humans differs from the diversity of escape mutants among ferrets? Perhaps a beta diversity-like metric would provide a more precise measure of the difference in mutational profiles between individuals and facilitate a quantitative comparison of the human and ferret results.

This is a great suggestion, and we thank the reviewers for not only suggesting this analysis but also proposing a specific statistic to use. We were unfamiliar with “beta diversity," but it indeed is an excellent statistic for this type of comparison. We have now used beta diversity to compare how variable the ferret sera are in terms of the sites where they exert selection relative to the two groups of human sera. Indeed, the ferret sera exhibit substantially lower beta diversity than both groups of human sera, using either the Shannon or Simpson index. We have added these results for the Simpson index (which seems most appropriate as it emphasizes the sites of strongest selection) to the Results section where we compare the ferret sera to the human sera.

2) While the results linking the natural diversity of influenza to the selected mutants are suggestive, the reviewers found claims about the relationship to global evolution difficult to fully justify. Specifically, given the small number of individuals in this study, it is unclear how we can classify a particular selective pressure being present in "only a subset" vs "many individuals." If such claims are to be made, a more explicit link needs to be made between the results presented in the manuscript the fraction of the global population that may exert a particular selective pressure on influenza. Also, while the results involving the 53 year-old patient suggest stability of selective pressure over short timescales, that stability over longer timescales may be important to explaining global patterns of influenza evolution. This additional work is not essential, but tempering this section may be called for.

The reviewers are correct that our study does not examine enough individuals to justify conclusions about what fraction of the human population exerts selection on a given mutation. We have removed the text about “only a subset" and “many individuals", and instead simply note that some sites that evolve in nature are selected by only some sera that we tested. We then added the following sentence to underscore that our sample size is too small to draw more than these anecdotal observations:

“Although the number of sera that we have characterized is too small to draw conclusions about population-level immunity, it is likely that evolutionary selection on a viral mutation depends both on how many individuals have immunity that is affected by the mutation and the magnitude of the antigenic effect in those individuals.”

3) The authors focus only on in vitro neutralization, which is fine. However, antibodies have many functions and in vitro neutralization doesn't always correlate well with protection in vivo. Other antibodies that are present (HA side binders, stalk binders, non-neutralizing antibodies, antibodies that bind HA but inhibit NA etc.) might not show much activity in an in vitro entry inhibition assay but might be highly relevant in vivo. There is nothing wrong with using an in vitro assay here and the take away points from the study are important. But the above needs to be discussed and it needs to be pointed out that many other factors might influence drift. Also please discuss why such high titers were present (>1:10,000), this will be a surprise to many readers.

These are good points. We have added a full paragraph to the Discussion that emphasizes how neutralizing anti-HA serum antibodies are an incomplete measure of immunity during actual human infections:

“It is important to note that our experiments only mapped in vitro immune selection exerted by neutralizing anti-HA antibodies in serum. During actual human infections, antibodies to HA can also exert selection by non-neutralizing mechanisms such as antibody-dependent cellular cytoxicity (He et al., 2016; Vandervan et al., 2018), activation of complement (Rattan et al., 2017), and inhibition of the viral neuraminidase (Kosik et al., 2019; Chen et al., 2019). In addition, antibodies in serum are mostly of the IgG class, but IgA antibodies predominate in the upper airway mucosa where human influenza virus typically replicates (Gould et al., 2017). But despite these caveats, many studies have shown that neutralizing serum activity is a strong correlate of protection against influenza (Hobson et al., 1972; Wagner et al., 1987; Krammer, 2019), so mapping selection from such serum provides an important if incomplete picture of the selection that human immunity imposes on the virus.”

We have also elaborated on why the serum neutralizing titers were so high. We explain that we pre-screened the human sera to find samples that had high activity in order to allow us to potently neutralize the viral library. As suggested by the reviewers, we have also added a sentence to clearly emphasize that this pre-screening means that the sera we examine has quite high titers:

“Notably, this pre-screening means that all the sera that we characterized had high neutralizing activity, with inhibitory concentrations 50% (IC50s) ranging from ~1:800 to ~1:3,000 (Supplementary File 2).”

4) The authors dilute the serum to approximately 1 IC90. By doing that, the effect of other antibodies present at lower titers that potentially target other sites is diminished or completely eliminated. The serum dilutions used should be stated and this should be discussed.

The reviewers are correct that this is an important point. We indeed performed the selections with high but not completely neutralizing serum concentrations (typically around the IC95). This is a crucial point to make clear, as the reviewers are correct that at these relatively high concentrations we are primarily seeing the effect of the antibodies that most strongly contribute to serum neutralization.

In fact, we already give the exact serum dilutions used in Supplementary file 3, and the exact percent of the library surviving each selection in Supplementary file 4. But the reviewers are correct that this information was somewhat buried. In addition, it's not easy to interpret the meaning of any particular serum dilution without also examining the overall potency of that serum to know how much neutralization occurred.

Therefore, we have made an addition that should make all of this transparently clear to the reader: in all of the figures showing serum mapping (Figures 2, 3, 4, and 5), we have added dashed vertical lines on the neutralization curve at the serum concentration used. These lines therefore explicitly show the serum dilution in the context of a curve showing the serum potency, so the reader can immediately understand not only the concentration but how high it is relative to the serum's neutralizing activity. In addition, we have added text to the figure legends explaining these lines, and text to the Results and Discussion emphasizing that these concentrations mean we primarily see selection from the antibodies that most strongly contribute to serum neutralization.

5) While IgG (which forms the majority of serum antibodies) protects the lung and is important for lower respiratory tract infections it is unlikely that it drives drift. Please add to the Discussion that the vast majority of influenza virus infections are restricted to the upper respiratory tract which is protected by IgA1.

This is a good point. We have added this caveat to the Discussion in a new paragraph, which discusses IgA versus IgG as well as the role of non-neutralizing antibodies. See our response to major point (3) above for the full text of the new paragraph we have added on these topics.

6) In a way it is concerning that mAb spiked into serum kind of 'takes over', diminishing the effect of the antibodies in serum in the set up used. This might be an artifact of the library system or the fact that the serum is strongly diluted. But this discrepancy needs to be pointed out because it suggests that one dominant antibody clonotype in serum could skew what the authors see when they perform their analysis. It really begs the question why serum neutralizing activity seems to become irrelevant as soon as highly neutralizing mAb is present? One possible explanation is, that the mAb actually sterically blocks binding of the dominant neutralizing antibody in serum? But then, mutants that escape the mAb should still be neutralized by serum antibody? Please clarify.

These are good points, and we have substantially expanded the text discussing Figure 7 to address these points. This expanded text focuses on two main points:

First, we point out the fact that our experiments identify the mutations with the largest relative effect on viral neutralization, but do not measure the absolute effect of any given mutation on antibody binding. In other words, once a single antibody dominates the neutralization (as the monoclonal antibody does in the spike-in experiments in Figure 7), only mutations in its epitope will have effects because escaping other antibodies no longer appreciably changes neutralization. But this fact does not mean that no antibodies are binding other sites, it just means that these other antibodies are sufficiently minor contributors to overall neutralization that mutating their epitopes have minimal effects.

Second, it is possible (as the reviewer suggests) that there could be non-additive effects such as steric hindrance, and we now explicitly discuss this. Importantly, even with these effects, we can still map escape mutations. The reason is that although mutants that escape the monoclonal antibody would then be bound by other serum antibodies, these other serum antibodies are less important for overall neutralization, so mutants that escape the monoclonal antibody will still be the most favored (enriched in our experiment), even if they are still partially neutralized by other serum antibodies. The reason is that we are using high but not completely saturating serum concentrations (usually around the IC95). If we used extremely high saturating amounts of serum, then we would expect no single mutation to ever escape if the serum was polyclonal.

7) Discussion section: "the importance of history is demonstrated by the lack of variation among ferrets that have been experimentally infected just once with a defined viral strain." This is exactly right, but then there should not be a claim anywhere in the paper that humans and ferrets are different, but rather that adult humans (with presumed multiple infections) and ferrets after a single infection are different. I agree that history rather than host biology seems the more likely explanation, but in the current study the two are confounded.

We agree that differences in exposure history are the most likely cause of the large differences between the human and ferret sera. The reviewers are correct that in some places we did not clearly articulate this fact, and instead simply referred to differences between ferrets and humans without clearly emphasizing that the ferrets were singly infected. We have therefore revised the manuscript to ensure that every place where the differences between the human and ferret sera are mentioned, we also clearly emphasize that the ferrets were infected just once. This includes an edit to the Abstract, several edits to the Results section on the ferret sera, and an edit to the Discussion.